# Analytic Bijections for Smooth and Interpretable Normalizing Flows

**Mathis Gerdes** [1] [2] [3]   **Miranda C. N. Cheng** [1] [4] [5]

## Abstract

A key challenge in normalizing flows is finding expressive invertible scalar bijections. Existing approaches face trade-offs: affine transformations are smooth and analytically invertible but lack expressivity; monotonic splines offer local control but are only piecewise smooth and act on bounded domains; residual flows achieve smoothness but need numerical inversion. We introduce three families of *analytic bijections* that are globally smooth ($C^\infty$), defined on all of $\mathbb{R}$, and analytically invertible in closed form, combining the favorable properties of prior approaches. Beyond serving as drop-in replacements in coupling flows, where they match or exceed spline performance, we develop *radial flows*: a novel architecture using direct parametrization that transforms the radial coordinate while preserving angular direction. Radial flows exhibit exceptional training stability, produce geometrically interpretable transformations, and on targets with radial structure can achieve comparable quality to coupling flows with $1000\times$ fewer parameters. We provide comprehensive evaluation on 1D and 2D benchmarks, and demonstrate applicability to higher-dimensional physics problems through experiments on $\phi^4$ lattice field theory, where our bijections outperform affine baselines and enable problem-specific designs that address mode collapse.

## 1. Introduction

Normalizing flows learn probability distributions by transforming a simple base density through invertible maps. In coupling (Dinh et al., 2015; 2017; Kingma & Dhariwal, 2018) and autoregressive (Papamakarios et al., 2017; Kingma et al., 2016; Huang et al., 2018) architectures, the choice of scalar bijection fundamentally constrains expressivity and training stability.

Existing scalar bijections face a tradeoff. *Affine transformations* (Dinh et al., 2017) are smooth and analytically invertible, but can only shift and scale. They lack the local expressivity needed to capture multimodal or heavy-tailed structure without many layers. *Monotonic splines* (Durkan et al., 2019) offer fine-grained local control through learnable knots, but are only $C^k$ for finite $k$ (not $C^\infty$) and only actively transform a bounded interval. Bijections in Gaussianization flows (Meng et al., 2020), *residual flows* (Chen et al., 2019; Behrmann et al., 2019; Ho et al., 2019) and related smooth constructions (Köhler et al., 2021) achieve global smoothness but require numerical root-finding for inversion. We focus here on discrete architectures, but note that *continuous normalizing flows* (Chen et al., 2018; Grathwohl et al., 2019) also define flexible and smooth transformations, however at the cost of numerically solving differential equations.

We introduce *analytic bijections* that resolve this trade-off: globally smooth ($C^\infty$), defined on all of $\mathbb{R}$, analytically invertible in closed form, and supporting both local deformations and global redistribution of probability mass. Our constructions derive from two principles: algebraic rational functions whose inverses reduce to solvable cubics, and conjugation with monotonic maps such as the $\sinh$ function.

Beyond improving coupling flows, we develop *radial flows*: a novel architecture that transforms the radius $r = \|\boldsymbol{x}\|$ while preserving angular direction. Unlike coupling flows with neural network conditioners, radial flow parameters can be learned directly, yielding exceptional training stability (learning rates an order of magnitude higher), geometric interpretability, and—on targets with radial structure—comparable quality with $1000\times$ fewer parameters.

In summary, our contributions are: (1) three parametric families of analytic bijections satisfying all desiderata simultane-

[1]Institute of Physics, University of Amsterdam, Netherlands [2]Department of Physics, Massachusetts Institute of Technology, Cambridge, MA, USA [3]The NSF AI Institute for Artificial Intelligence and Fundamental Interactions [4]Korteweg-de Vries Institute for Mathematics, University of Amsterdam, Netherlands [5]Institute for Mathematics, Academia Sinica, Taiwan. Correspondence to: Mathis Gerdes <mgerdes@mit.edu>, Miranda C. N. Cheng <mcheng@uva.nl>.

*Proceedings of the 43rd International Conference on Machine Learning*, Seoul, South Korea. PMLR 306, 2026. Copyright 2026 by the author(s).

ously; (2) radial flow architectures with direct parametrization; (3) numerical evaluation demonstrating competitive performance with qualitative smoothness advantages.

## 2. Background

**Normalizing flows.** Normalizing flows (Tabak & Vanden-Eijnden, 2010; Papamakarios et al., 2021; Kobyzev et al., 2021) define an invertible transformation $f_\theta : \mathbb{R}^d \to \mathbb{R}^d$ from a base distribution $\rho$ (typically Gaussian) to the target. The model density follows from the change of variables:

$$q_\theta(x) = \rho(f_\theta^{-1}(x)) \left| \det J_{f_\theta}(f_\theta^{-1}(x)) \right|^{-1} , \qquad (1)$$

where $J_f(x) = \partial f / \partial x$ denotes the Jacobian of $f$ evaluated at $x$. Flows are typically constructed as compositions $f_\theta = f^{(L)} \circ \cdots \circ f^{(1)}$ of simpler bijective layers, where each layer $f^{(\ell)}$ has an efficiently computable log-determinant Jacobian.

**Coupling layers.** Coupling layers (Dinh et al., 2017) partition coordinates into passive $x_{\text{pass}}$ and active $x_{\text{act}}$ subsets:

$$y_{\text{pass}} = x_{\text{pass}}, \quad y_{\text{act}} = h(x_{\text{act}}; \theta(x_{\text{pass}})), \qquad (2)$$

where $h$ is a scalar bijection applied element-wise, and $\theta = g_\phi(x_{\text{pass}})$ are the bijection parameters output by the conditioner network. The triangular Jacobian yields efficient log-determinant computation: $\log|\det J| = \sum_i \log |h'(x_{\text{act},i})|$.

## 3. Analytic Bijections

The choice of scalar bijection $h$ critically affects expressivity. We seek bijections satisfying five desiderata: (1) global smoothness ($C^\infty$); (2) global domain ($\mathbb{R}$); (3) analytic closed-form invertibility; (4) tractable Jacobian; (5) expressive parametrization supporting local (and thus non-linear) deformations. Table 1 shows how existing methods fall short; our constructions achieve all five.

*Table 1.* Comparison of scalar bijection properties. Our analytic bijections satisfy all desiderata.

| Method | Smooth | Global | Inverse | Local |
|---|---|---|---|---|
| Affine | ✓ | ✓ | ✓ | ✗ |
| Splines | $C^k$ only | ✗ | ✓ | ✓ |
| Residual | ✓ | ✓ | ✗ | ✓ |
| *This work* | | | | |
| Cubic rational | ✓ | ✓ | ✓ | ✓ |
| Sinh conj. | ✓ | ✓ | ✓ | ✓ |
| Cubic conj. | ✓ | ✓ | ✓ | ✓ |

In coupling layers, *affine* maps $h(x) = e^s x + t$ of Real NVP (Dinh et al., 2017) are simple and stable but limited

to global scaling and shifting. *Monotonic splines* (Durkan et al., 2019; Dolatabadi et al., 2020) use piecewise rational-quadratic functions for local control but actively transform only bounded intervals. While variants (Hong & Chun, 2023) can achieve $C^k$ smoothness for finite $k$, they lack $C^\infty$ regularity.

### 3.1. Construction Methods

We present two distinct construction principles for scalar bijections that meet the five desiderata: algebraic rational functions whose inverses reduce to a solvable cubic, and conjugation with monotonic maps.

**Algebraic rational functions.** We seek algebraic bijections of the form $h(x) = x + g(x)$, where $g(x)$ is a perturbation vanishing as $|x| \to \infty$. We require $h(x) \to x$ at large $|x|$ so the bijection acts as a local deformation while preserving tail behavior; together with linear growth at infinity this also ensures the transformed distribution inherits the prior's support and tails, an important property for stable training (see also Appendix B.3). Taking $g(x) = n(x)/d(x)$ with $n$ and $d$ polynomials leads to the following constraints. First, $d$ must have no real roots, requiring it to be of even degree. Second, the degree of $n$ must be strictly less than that of $d$ so that $g(x) \to 0$ as $|x| \to \infty$. For $\deg d \geq 4$, clearing denominators yields a quintic or higher-degree equation with no closed-form solution (Abel–Ruffini theorem). Degree 0 yields only affine maps. Thus $\deg d = 2$ is the unique non-trivial choice, yielding a *cubic rational* bijection[1]:

$$h(x) = x + \frac{c_1 x + c_2}{c_3 x^2 + c_4 x + c_5} . \qquad (3)$$

Bijectivity constrains the parameters $c_i$; a tractable specialization with explicit constraints is given in equation (6).

**Conjugation with monotonic functions.** Given a strictly monotonic $g : \mathbb{R} \to \mathbb{R}$ with known inverse, we define

$$h(x) = g^{-1}(g(x) + \delta). \qquad (4)$$

This is bijective for any $\delta \in \mathbb{R}$ with derivative $h'(x) = g'(x)/g'(h(x))$. To get $h(x) \to x$ as $|x| \to \infty$, we require $g$ to be superlinear.[2] Additional parameters increase the expressivity of the bijection:

$$h(x) = g_{\sigma,\gamma}^{-1}(e^\mu(e^\nu g_{\sigma,\gamma}(x) + \delta)), \qquad (5)$$

where $g_{\sigma,\gamma}(x) = g((x - \gamma)/\sigma)$. The parameters $\mu, \nu$ create global shifts: distant points are displaced by a constant offset even though $h'(x) \to 1$.

---

[1] So named because clearing denominators gives a cubic equation in $x$ and $y = h(x)$

[2] Taylor expanding $g^{-1}$ around $g(x)$: $h(x) = x + \delta/g'(x) + O(\delta^2)$. When $\delta \neq 0$, this approaches $x$ if and only if $g'(x) \to \infty$, i.e., $g$ grows faster than linear.

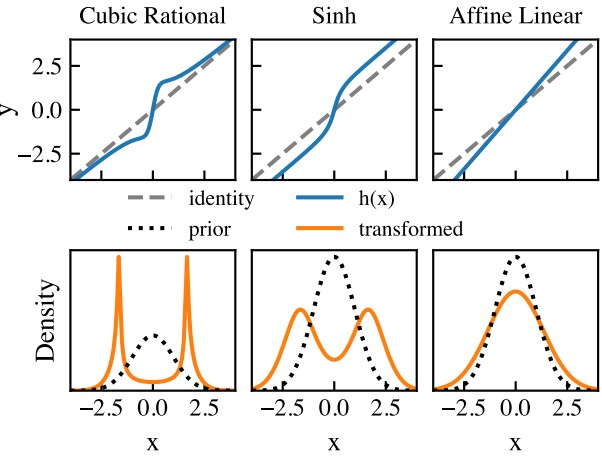

*Figure 1.* Three bijection behaviors. *Left*: Local deformation (cubic rational) creates bimodal structure while $h(x) \to x$ for large $|x|$. *Middle*: Global shift (sinh) displaces distant points by constant offset. *Right*: Uniform scaling (affine, for comparison).

### 3.2. Three Parametric Families

**Cubic rational.** The rational form

$$h(x) = x + \frac{\lambda(x - \gamma)}{1 + (x - \gamma)^2/\sigma^2} \qquad (6)$$

satisfies the requirements for the algebraic construction: clearing denominators yields a cubic in $x$ solvable via Cardano's formula. Bijectivity requires $-1 < \lambda < 8$ and $\sigma > 0$ (see Appendix A).

The expression is purely algebraic (no transcendental functions), yielding strictly local deformations in the sense that $h(x) \to x$ as $|x| \to \infty$. Inverse requires Cardano's formula.

**Sinh conjugation.** Using $g = \sinh$ in the conjugation construction:

$$h(x) = \sigma \cdot \operatorname{arcsinh}(e^{\mu}(e^{\nu}\sinh\left(\tfrac{x-\gamma}{\sigma}\right) + \delta)) + \gamma. \qquad (7)$$

This supports both local deformations (via $\delta$) and global shift via $\mu, \nu$ (see Figure 1).[3]

**Cubic conjugation.** Using $g(x) = ax + bx^3$ ($a, b > 0$):

$$h(x) = g^{-1}(g(x - \gamma) + \delta) + \gamma. \qquad (8)$$

Both forward and inverse require Cardano's formula. Purely algebraic like cubic rational, but follows the conjugation structure. No global shift parameters $\mu, \nu$ as they can be absorbed into $a, b, \delta$.

---

[3]Note that the parameter $\nu$ could be absorbed into $\mu$ and $\delta$, but with the above parametrization the inverse and forward maps are symmetric (switch $\mu$ and $\nu$, flip sign of $\delta, \mu, \nu$).

### 3.3. Local and Global Deformations

Figure 1 visualizes the distinction between purely local deformations compared to a global shift and global scaling. All analytic bijections support local deformations, which require compensating a "stretching" in some region of input space with a "compressing" in a nearby region. The sinh conjugation supports a global shift, which is distinct from the global uniform stretching of affine linear maps. The latter cannot perform local deformations, thus always affecting the entire distribution. Spline flows, when extended with the identity map outside their active domain, are restricted to purely local deformations. Composing a chain of different bijections allows for a combination of their behaviors.

### 3.4. Stacking and Application in Coupling Flows

While expressivity of spline flows is increased by tuning the number of knots, the primary way to do this with our bijections is to compose multiple independently parameterized copies into one composite scalar bijection. We refer to this composition depth as the *stack count* $N$; each layer in the stack is a fresh instance of the same bijection family with its own trainable parameters.

These stacks serve as a drop-in replacement for spline flows or the affine linear map, for example in autoregressive or coupling flow architectures, at memory cost and runtime comparable to splines (see Appendix B.8). Other applications include radial flows, as presented in Section 4, and problem-specific transformations such as zero-mode bijections in lattice field theory to prevent mode collapse (see Section 5.5).

**Training stability.** Constrained parameter ranges must be enforced by differentiable transformation of raw trainable parameters or network outputs (in the case of coupling layers). We use softplus for positivity ($\sigma, a, b > 0$) and sigmoid for bounded intervals ($\lambda$). Especially in deep coupling flows, careful initialization is important (Andrade, 2024; Hong et al., 2023; Koehler et al., 2021). Suppressing the initialization scale of the conditioner's final layer improves training stability, as it prevents initial parameters that push bijections toward extreme transformations. See Appendix C.4 for details.

## 4. Radial Flows

We now develop a novel architecture that leverages the expressivity of our bijections: transforming the radial coordinate, optionally conditioned on the angular coordinates, while preserving angular direction.

Any point $\boldsymbol{x} \in \mathbb{R}^n \setminus \{\mathbf{0}\}$ decomposes as $\boldsymbol{x} = r\hat{\boldsymbol{x}}$ with $r = \|\boldsymbol{x}\|$ and $\hat{\boldsymbol{x}} = \boldsymbol{x}/r$. A *radial transformation* applies a

scalar bijection $f : \mathbb{R}_{\geq 0} \to \mathbb{R}_{\geq 0}$ to the radius:

$$g(\boldsymbol{x}) = \frac{f(\|\boldsymbol{x}\|)}{\|\boldsymbol{x}\|} \boldsymbol{x}. \tag{9}$$

The log-Jacobian has a simple closed form:

$$\log|\det J_g| = \log|f'(r)| + (n-1)\log\left|\frac{f(r)}{r}\right|. \tag{10}$$

We enforce $f(0) = 0$ by using $f(r) = \tilde{f}(r) - \tilde{f}(0)$ with $\tilde{f}$ unconstrained, preserving bijectivity and smoothness.

### 4.1. Radial Bijections

Each radial layer has a learnable center $\boldsymbol{c} \in \mathbb{R}^n$, optional per-dimension scaling $\boldsymbol{s} \in \mathbb{R}^n_{>0}$, and a scalar bijection $f$ (with constraint $f(0) = 0$). The transformation is applied relative to the center:

$$g(\boldsymbol{x}) = \boldsymbol{c} + (f(r)(\boldsymbol{x} - \boldsymbol{c}))/r, \tag{11}$$

where $r = \|\boldsymbol{s} \odot (\boldsymbol{x} - \boldsymbol{c})\|$.

Expressivity comes from stacking: defining $f$ as a chain of analytic scalar bijections per center increases radial expressivity; multiple centers enable angular redistribution—mass can shift between *rays* (half-lines of constant direction $\hat{\boldsymbol{x}}$ emanating from the center) through composition, see discussion below.

### 4.2. Angular Dependence

For direction-dependent transformations, we allow the radial bijection to depend on the angle coordinates: $r' = f(r, \hat{\boldsymbol{x}})$. The dependence on $\hat{\boldsymbol{x}}$ may be defined by an arbitrary neural network, as in coupling flows[4]. The Jacobian of equation (10) remains valid.

In 2D, where $\hat{\boldsymbol{x}}$ is equivalently expressed as the angle $\phi = \operatorname{atan2}(x_2, x_1)$, we can parametrize bijection parameters by a truncated Fourier series:

$$\theta_j(\phi) = a_{j,0} + \sum_{k=1}^{K}[a_{j,k}\cos(k\phi) + b_{j,k}\sin(k\phi)]. \tag{12}$$

This could be extended to higher dimensions using spherical harmonics.

**Regularity at the origin.** The unit vector $\hat{\boldsymbol{x}} = \boldsymbol{x}/\|\boldsymbol{x}\|$ is undefined at the origin, introducing a coordinate singularity in the polar decomposition—distinct from the smoothness

of the scalar bijections themselves. Since the transformation is defined as $g(\boldsymbol{x}) = (f(r, \hat{\boldsymbol{x}})/r)\boldsymbol{x}$, its linear approximation near the origin is

$$g(\boldsymbol{x}) \approx \partial_r f(0, \hat{\boldsymbol{x}}) \cdot \boldsymbol{x}. \tag{13}$$

For $g$ to be differentiable at the origin, this must be independent of the approach direction $\hat{\boldsymbol{x}}$; otherwise, the Jacobian is ill-defined. Smoothness thus requires $\partial_r f(0, \hat{\boldsymbol{x}}) = c$ for an $\hat{\boldsymbol{x}}$-independent constant $c$. This constraint can be enforced by composing with a final corrective bijection whose parameters are fixed[5] given $f'(0; \hat{\boldsymbol{x}})$. In our experiments, we found that enforcing this constraint—especially for few-layer flows—does not improve training performance. Whether to do so is thus application-dependent and can even be enforced post-training. For our qualitative evaluation, not enforcing the constraint has the benefit of clearly visualizing the origin location (see Figure 6).

### 4.3. Advantages & Trade-Offs

**Training stability.** In our experiments, radial flows train stably at learning rates of $10^{-2}$, an order of magnitude higher than coupling flows (see Table 5 in the appendix). Their geometric simplicity prevents the extreme Jacobians that destabilize coupling layers.

**Interpretability.** Each layer's action is geometrically intuitive: stretching or compressing radially around its center. Angle dependence can be introduced in a controlled and inspectable manner via Fourier series.

**Parameter efficiency.** On targets with radial structure, radial flows can achieve comparable quality to coupling flows with orders of magnitude fewer parameters (Section 5.3).

**Geometric constraints.** A single radial layer preserves rays: probability mass can be redistributed *along* rays from the center but not *between* them. For a single-center flow to approximate a target, the target's angular mass distribution must approximately match the prior's for some center point. Stacking multiple radial layers with different learned centers mitigates this constraint. However, the number of layers required may scale adversely with dimension. In practice, radial flows may also be combined with normalizing flows on the sphere.

### 4.4. Related Work

Simple radial transformations were introduced by Rezende & Mohamed (2015) for variational inference, using maps

---

[4]In fact, angle-dependent radial flows can be seen as a special case of coupling flows in spherical coordinates, with the radius as only active coordinate. This connection may be further pursued in future work by updating also subsets of angle coordinates conditioned on the radius and frozen angles.

[5]For example, the derivative can be set to 1 by adding a final sinh bijection with $\gamma = \sigma \cdot \operatorname{arcsinh}(1/f'(0; \hat{\boldsymbol{x}}))$, $\delta = 1/f'(0; \hat{\boldsymbol{x}}) - f'(0; \hat{\boldsymbol{x}})$ and $\mu = \nu = 0$, since with this choice, $h'_{\sinh}(0; \hat{\boldsymbol{x}}) = 1/f'(0; \hat{\boldsymbol{x}})$.

of the form $r' = r + r\beta/(\alpha + r)$, with $\alpha, \beta \in \mathbb{R}$. Our radial flows build on this foundation substantially: (1) we replace the single simple transformation with our analytic bijections; (2) we use multiple learnable centers and basis scalings; (3) we introduce angular dependence.

# 5. Numerical Experiments

We evaluate our analytic bijections and flow architectures across four settings: one-dimensional density estimation, two-dimensional coupling and radial flows, and a physics application to lattice field theory. We focus on discrete flows with closed-form inverses; continuous normalizing flows and residual flows require numerical integration or inversion and represent different tradeoffs outside our scope.

**Loss functions.** We use standard loss functions derived from the Kullback-Leiber (KL) divergence

$$D_{\mathrm{KL}}(p_1\|p_2) = \int p_1(x) \log(p_1(x)/p_2(x)) \, \mathrm{d}x \ . \quad (14)$$

When samples from the target distribution $p$ are available, we use the *forward* KL divergence $D_{\mathrm{KL}}(p\|q_\theta) = -H_p + \mathcal{L}$. The entropy $H_p$ is constant with respect to model parameters and thus omitted during training, yielding as training loss the negative log-likelihood $\mathcal{L} = -\mathbb{E}_{x\sim p} \log q_\theta(x)$.

In scientific contexts the target is often given in terms of an unnormalized density $\tilde{p} = pZ_p$ with unknown normalization constant $Z_p$. In this case, we use the *reverse* KL divergence $D_{\mathrm{KL}}(q_\theta\|p) = \mathcal{L} + \log Z_p$. We drop the unknown partition function $Z_p$, minimizing $\mathcal{L} = \mathbb{E}_{x\sim q_\theta}[\log q_\theta(x) - \log \tilde{p}(x)]$.

For low-dimensional benchmarks, we numerically estimate the partition function $Z_p$ and entropy $H_p$, enabling visualization of the absolute KL divergence with reference to the optimum at $D_{\mathrm{KL}} = 0$.

## 5.1. 1D Stack

We evaluate our bijections in isolation by training 1D normalizing flows on a synthetic target distribution[6] with reverse KL divergence. Flows are constructed by stacking $N \in \{3, 9, 27, 128, 256\}$ scalar bijections. These compositions also serve as building block for higher dimensional flows like coupling and radial flows, as discussed in the next sections.

Figure 2 shows that all three bijection types improve with increasing depth $N$. All results are averaged over training with 6 different seeds; error bars show one standard deviation. At $N = 27$, cubic conjugation achieves effective

---

[6]Manually chosen to obtain a non-trivial multi-modal structure: $\log p(x) = a_1 \sin(\omega_1 x)e^{-\gamma_1 x^2} + a_2 \cos(\omega_2 x) + a_4 x^4$, with $a_1 = 1, a_2 = 2, \omega_1 = 5 \, \omega_2 = 10, a_4 = -0.2, \gamma_1 = 5$.

sample size[7] (ESS) $\approx 99\%$ and forward $D_{\mathrm{KL}} \approx 3.5 \times 10^{-3}$. Figure 12 in the appendix shows the training dynamics for each bijection type.

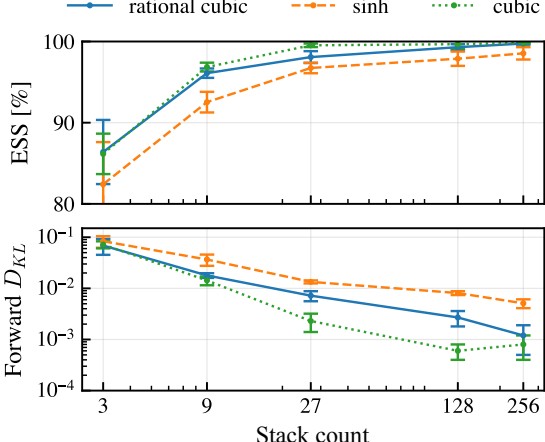

*Figure 2.* 1D flow performance vs. number of stacked bijections. Top: Effective Sample Size (higher is better). Bottom: Forward KL divergence (lower is better). All bijection types show monotonic improvement with depth, with cubic conjugation performing best.

## 5.2. Coupling Layer Integration

We train coupling flows (12 layers, ResNet conditioners) on a 2D spiral distribution using maximum likelihood loss, comparing affine, rational quadratic spline (8 knots), and our bijections with stack count $N \in \{1, 3, 9, 27\}$ per layer.

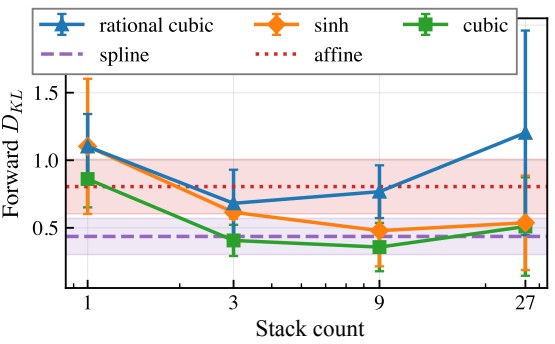

*Figure 3.* Forward KL for 2D coupling flows. Our bijections (markers) improve with depth, with cubic outperforming both affine and spline baselines (horizontal lines) at $N = 9$. For larger $N$, increasing expressivity decreases training stability and increases variance in final performance.

Figure 3 shows final forward KL divergence. All results are averaged over training with 6 different seeds; error bars show one standard deviation. At $N = 9$, the cubic conjugation bijection achieves the best final performance

---

[7]The effective sample size ESS = $(\sum_i w_i)^2 / \sum_i w_i^2$ where $w_i = p(x_i)/q_\theta(x_i)$ measures sample quality.

($D_{\mathrm{KL}} \approx 0.35$), outperforming both affine ($D_{\mathrm{KL}} \approx 0.8$) and spline ($D_{\mathrm{KL}} \approx 0.45$) baselines. This validates our bijections as effective drop-in replacements in standard coupling architectures. Training dynamics are shown in Figure 15 in the appendix.

### 5.3. Radial Flow Evaluation

**Stack of radial flows.** We train angle-independent radial flows on the 2D spiral, varying the number of centers $L \in \{5, 10, 20, 40\}$ and stacked bijections per center $N \in \{1, 2, 8, 16, 32\}$. Figure 4 shows forward KL divergence across configurations. Performance improves with more centers and larger stack count. With too large a stack count, training performance deteriorates due to training instability, which may be addressed by tuning learning rate and other hyperparameters, which were held fixed across configurations here.

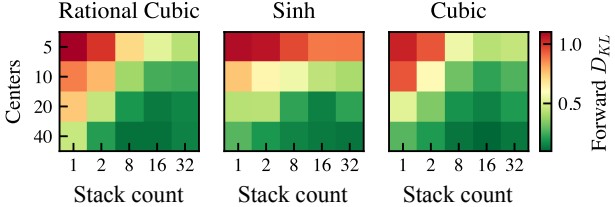

*Figure 4.* Forward KL divergence for radial flows vs. number of centers and stacked bijections. Performance improves systematically with both hyperparameters. Results averaged over 6 independent training runs with different random seeds.

**Interpretable Fourier radial flows.** Next we consider a single layer of the angle-dependent radial flow introduced in Section 4.2 with a stack of $N = 9$ sinh bijections, trained on the spiral target. Figure 5 shows how the learned density improves with the number of Fourier terms (with $K = 0, 1, 2, 3$). With no angle dependence ($K = 0$), the flow approximates the target's radial structure as concentric circles. Adding angular modes progressively improves the spiral detail, reaching high fidelity with just 319 parameters at $K = 3$ ($2K + 1 = 7$ terms per bijection parameter). Both knobs (angular terms and stack count) saturate beyond modest values: with $N = 9$ sinh copies, the test NLLs at $K \in \{0, 1, 2, 3\}$ are $\{-0.09, -0.61, -0.69, -0.74\}$, and further increasing depth to $N = 32$ at $K = 2$ reaches $-0.79$. Unlike coupling layers, the transformation can be interpreted and inspected as a single angle-dependent bijection $f(r, \phi)$ (see Figure 18 in the appendix for a direct visualization).

**Qualitative smoothness.** Figure 6 compares learned densities during training on the spiral target of the Real NVP (affine) coupling flow of Section 5.2 against a single-layer Fourier radial flow with $N = 32$ sinh-bijections and $K = 2$.

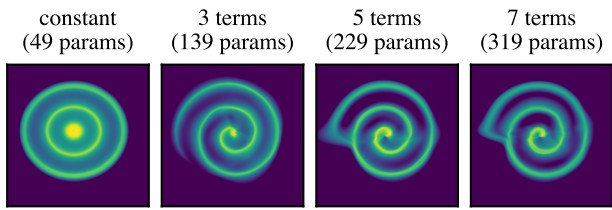

*Figure 5.* Fourier radial flows with increasing angular terms on a spiral target. Even 49 parameters (constant term only) capture radial structure; 319 parameters (7 terms) achieve high fidelity.

Both architectures fit the target well: the Fourier radial flow attains test NLL $-0.79$ versus $-0.52$ for the RealNVP baseline, with the radial flow using roughly three orders of magnitude fewer parameters. Despite the comparable qualitative fit, the coupling flow exhibits characteristic "folding" artifacts—thin lines of high probability from iterative axis-aligned transformations (see Figure 16 in the appendix).

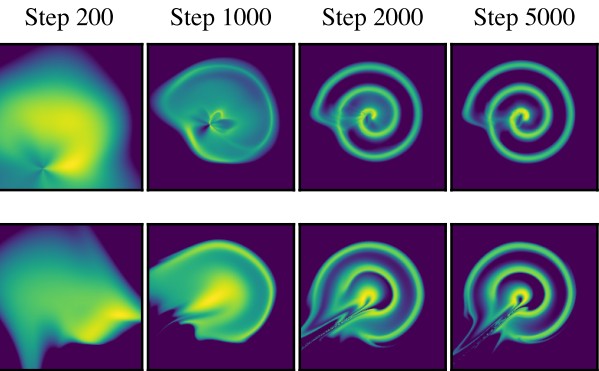

*Figure 6.* Density evolution during training. Top: Radial Fourier flow. Bottom: Coupling flow (Real NVP). Both achieve similar $D_{\mathrm{KL}}$, but radial flows produce smoother densities without compression artifacts. The coupling flow was trained with lower learning rate to avoid instability (see Section C.5 in the appendix).

For the radial flow, we furthermore initialized the center at $(-0.5, -1)$. Training successfully moves the center towards the spiral center, confirming that radial flows can learn the center required to maximize angular uniformity, and it need not be tuned manually. Note that initially the center is visible as a distinct singular point, as we do not enforce $\partial_r f(0; \hat{\boldsymbol{x}}) = \mathrm{const}$ here (see Section 4.2).

**Source distributions.** Figure 7 compares the inverse mapping of target samples for the coupling and radial flows of the previous paragraph. By construction, the radial flow preserves a clean radial organization in the source: inner spiral points (blue) map near the origin, while outer points (red) lie at larger radii. In contrast, the affine coupling flow mixes these regions through alternating coordinate-wise transformations, producing the folding artifacts visible in Figure 6 (see Appendix Figure 16; a spline-coupling variant

alleviates but does not eliminate these axis-aligned patterns).

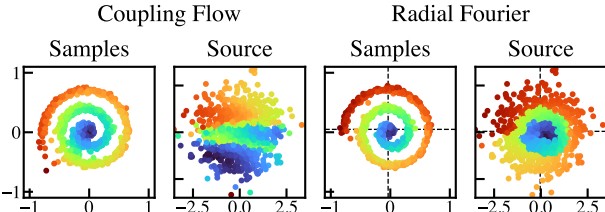

*Figure 7.* Source distributions (inverse images) colored by position in the target spiral over samples from trained models. The radial flow (right) preserves structure, mapping inner points (blue) to the origin; the coupling flow (left) mixes the radial structure.

**Multimodal targets.** To demonstrate that a stack of radial flows with independent learnable centers can approximate multimodal targets violating angular uniformity, we train on a 5-component Gaussian mixture arranged in a circle. We use 32 centers, and train a standard coupling flow (as above), an angle independent radial flow, and one angle-dependent radial flows with 5 Fourier terms ($K = 2$). In all cases we use a stack of 12 cubic bijections, and train with negative log-likelihood loss. Figure 8 shows that the pure radial flow (1.6k parameters) achieves the best visual fidelity, while the coupling flow (2,311k parameters) exhibits spiky artifacts.

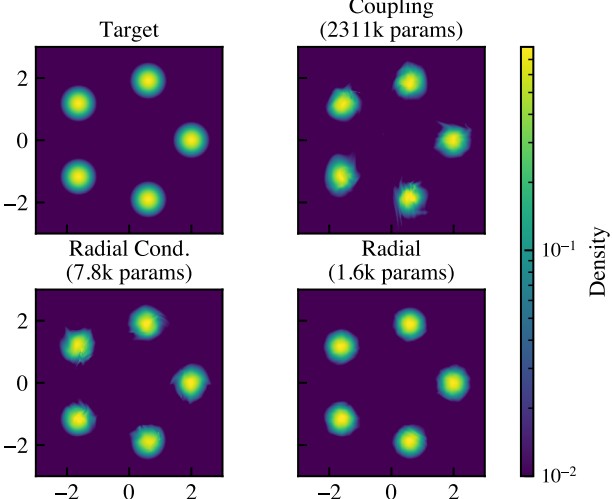

*Figure 8.* Learned densities for 2D Gaussian mixture. Radial flow best reproduces smooth circular blobs; coupling and to some extent angle-dependent radial flows show more significant artifacts. All use cubic bijections.

### 5.4. Density Estimation Benchmarks

To further validate our analytic bijections as drop-in replacements, we evaluate them on two standard density-estimation benchmarks: image density estimation on CIFAR10 and tabular density estimation on the UCI (see Papamakarios

et al. (2017)). Across both benchmarks, all variants share the same per-experiment architecture and hyperparameters; only the scalar bijection inside the coupling layer changes. Architecture and training details for both benchmarks are in Section C.5 and Section C.6.

**CIFAR10.** We train a multi-scale Real NVP-style flow (Dinh et al., 2017), replacing the affine bijection of each coupling layer by a stack of 8 analytic bijections followed by a trailing affine ("RealNVP+"). Our reimplementation departs from the original RealNVP in minor ways (e.g. ActNorm in place of batch normalization, GELU conditioner); see Section C.6 for details. Table 2 reports test bits per dimension (lower is better). All three analytic variants improve over the published affine RealNVP baseline by roughly 0.12 bpd at matched architecture, confirming that the analytic bijections deliver consistent gains as drop-in replacements for the affine coupling.

*Table 2.* CIFAR10 test bits per dimension (lower is better). All "RealNVP+" variants share the same multi-scale architecture and differ only in the scalar bijection used inside each coupling layer.

| Model | BPD |
|---|---|
| RealNVP (Dinh et al., 2017) | 3.49 |
| RealNVP+ (cubic rational) | **3.36** |
| RealNVP+ (sinh conjugation) | 3.37 |
| RealNVP+ (cubic conjugation) | 3.37 |

**UCI tabular benchmarks.** We follow the coupling architecture of Durkan et al. (2019) (the coupling variant of their rational-quadratic neural spline flow, RQ-NSF(C)), reimplemented in JAX, and vary only the scalar transformer inside each coupling layer. We compare three variants: *spline* (our reproduction of RQ-NSF(C)), *sinh* (a stack of sinh conjugation bijections inside each coupling layer), and *spline+* (a stack of sinh conjugation bijections followed by a rational-quadratic spline). Per-dataset hyperparameters are held fixed across our three variants but differ in minor ways from Durkan et al. (2019); the reproduced *spline* baseline therefore serves as a controlled within-architecture reference. See Section C.6 for details. Table 3 reports test log-likelihood (higher is better).

Combining sinh and spline (*spline+*) generally improves over the spline-only baseline, matching or exceeding the published RQ-NSF(C) numbers on POWER and BSDS300. The exceptions are HEPMASS and MINIBOONE, where *spline+* underperforms the reproduced *spline*—consistent with their small-dataset, overfitting-prone regime, in which added expressivity does not help; on MINIBOONE the pure *sinh* variant is in fact the strongest of all methods, slightly exceeding RQ-NSF(C). The pure *sinh* variant is competitive with *spline* across all datasets but not consistently better, indicating that splines and analytic bijections capture com-

*Table 3.* Test log-likelihood on UCI tabular datasets (higher is better). RQ-NSF(C) numbers are reproduced from Durkan et al. (2019); *spline* is our within-framework reproduction. All our variants share per-dataset hyperparameters; only the scalar bijection inside each coupling layer changes. Bold marks the best result among our three variants on each dataset.

| Method | POWER | GAS | HEPMASS | MINIBOONE | BSDS300 |
|---|---|---|---|---|---|
| RQ-NSF(C) | $0.64 \pm 0.01$ | $13.09 \pm 0.02$ | $-14.75 \pm 0.03$ | $-9.67 \pm 0.47$ | $157.54 \pm 0.28$ |
| spline (ours) | $0.62 \pm 0.04$ | $12.82 \pm 0.05$ | $\mathbf{-14.80 \pm 0.36}$ | $-9.68 \pm 0.22$ | $157.45 \pm 0.12$ |
| sinh | $0.62 \pm 0.03$ | $12.75 \pm 0.04$ | $-15.12 \pm 0.52$ | $\mathbf{-9.58 \pm 0.23}$ | $157.56 \pm 0.29$ |
| spline+ | $\mathbf{0.64 \pm 0.01}$ | $\mathbf{12.92 \pm 0.05}$ | $-14.95 \pm 0.17$ | $-10.35 \pm 0.62$ | $\mathbf{157.57 \pm 0.24}$ |

plementary structure, which the *spline+* hybrid exploits.

### 5.5. Physics Application: Lattice Field Theory

To validate scaling to higher dimensions, we apply our bijections to $\phi^4$ lattice field theory on a $L \times L = 20 \times 20$ lattice, a scientific domain in which normalizing flows have emerged as tools for accelerating Monte Carlo sampling (Albergo et al., 2019; Kanwar, 2024; Cheng & Stratikopoulou, 2026). The target density $p(\phi) \propto \exp(-S[\phi])$ is defined by:

$$S[\phi] = \sum_x \left[ \sum_{\mu=1}^{2} (\phi_{x+\hat{\mu}} - \phi_x)^2 + m^2 \phi_x^2 + \lambda \phi_x^4 \right], \quad (15)$$

where $x$ indexes lattice sites and the sum over $\hat{\mu}$ captures nearest-neighbor interactions with periodic boundaries. We fix $m^2 = -4$, creating a double-well potential at each site, and vary the parameter (quartic coupling constant) $\lambda$.

**Improving coupling flows.** For larger $\lambda$, the distribution is effectively unimodal. We use $\lambda = 4.807$, yielding a correlation length of $\xi \approx L/4$. This regime tests whether our analytic bijections can improve standard coupling architectures when replacing the simple affine map of Real NVP. We train a 12-layer coupling flow using checkerboard masking and small convolutional networks (2 hidden layers, 16 channels, $3 \times 3$ kernels). To effectively learn the target distribution's power spectrum, we include a learnable, rotation-symmetric *Fourier scaling* layer $\tilde{\phi}_k \mapsto e^{s_{|k|}} \tilde{\phi}_k$ before the coupling layers. The trainable parameters $s_{|k|}$ are shared across all Fourier modes with the same momentum magnitude $|k|$, consistent with the rotation and reflection symmetries of the lattice.

The affine-only Real NVP baseline reimplements the architecture of Albergo et al. (2019). We compare it as well as a quadratic rational spline variant against three variants that stack 8 analytic bijections—either sinh, cubic, or cubic rational—followed by a final affine transformation per layer. Figure 9 shows that all three analytic bijections achieve consistently higher effective sample size than the spline and affine baselines, with final ESS ordering cubic rational (39.66%) > cubic (38.85%) > sinh (38.51%) > spline (34.34%) > affine (31.85%). This demonstrates that our bijections provide meaningful improvements even in 400 dimensions with genuine physical structure, and that the

smoothness advantage over splines persists when scaled beyond toy targets (see also Appendix B.3 for a controlled smoothness comparison).

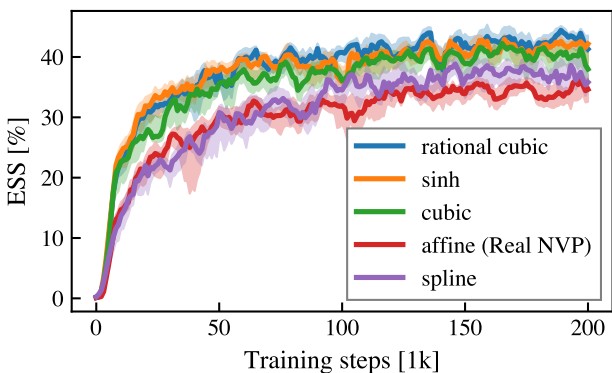

*Figure 9.* Effective sample size during training for $\phi^4$ theory. Analytic bijections (cubic, sinh, cubic rational) achieve higher ESS than both the spline and affine baselines. Median with interquartile range (shaded) over 3 runs, smoothed with 5-step moving average.

**Preventing mode collapse.** For smaller $\lambda$, the density becomes bimodal with a $\mathbb{Z}_2$ symmetry $p(\phi) = p(-\phi)$. Standard training with reverse KL divergence $D_{\mathrm{KL}}(q\|p)$ suffers from mode collapse: the loss can be minimized by covering only one of the two modes, which standard metrics like ESS (evaluated on samples from $q$) fail to detect.

We address this with a *zero-mode bijection*: a scalar bijection on the magnitude $|\tilde{\phi}_0|$ of the zero-frequency Fourier mode, with $f(0) = 0$ enforced as in radial flows. Since $\tilde{\phi}_0 \propto M = L^{-2} \sum_x \phi_x$ is the average magnetization (the $\mathbb{Z}_2$ order parameter), transforming only the magnitude preserves exact symmetry. We use a stack of 8 cubic conjugation bijections. Figure 10 shows the resulting magnetization $M$. Naive training achieves high ESS (90%) but collapses to a single mode. Pretraining the zero-mode bijection (with coupling layers frozen) ensures balanced sampling of both modes, which is maintained when full training resumes. For further details see Appendix B.7.

## 6. Conclusion

We introduced *analytic bijections*—cubic rational, sinh, and cubic conjugation—that combine global smoothness,

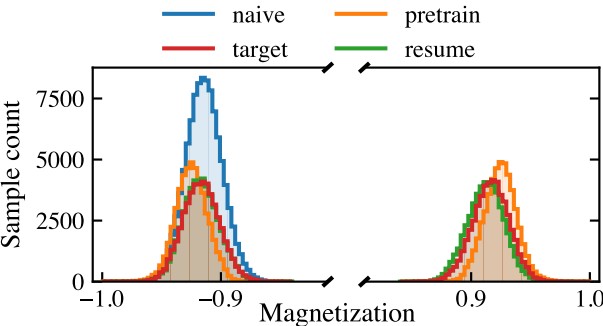

*Figure 10.* Magnetization histograms for $\phi^4$ in the bimodal regime. Naive training collapses to one mode. Pretraining the zero-mode bijection captures both modes, closely matching MCMC ground truth.

unbounded domain, closed-form inverses, and expressive parametrization supporting both local deformations and global redistribution of probability mass. These bijections serve as drop-in replacements for affine or spline transformations in coupling flows, achieving competitive or superior performance while guaranteeing smooth learned densities[8].

We further leverage the analytic bijections to develop *radial flows*: a novel architecture based on polar decomposition that transforms radius while preserving angular direction. Radial flows allow for direct parametrization, exhibiting exceptional training stability and geometric interpretability. On 2D benchmarks, radial flows match coupling flow quality with orders of magnitude fewer parameters and avoid the axis-aligned artifacts that affine coupling flows tend to produce on geometrically structured targets. The Fourier variant demonstrates that complex distributions can be captured with as few as 50–300 learned parameters, yielding a high level of interpretability and training stability.

On standard density-estimation benchmarks (Section 5.4), analytic bijections used as drop-in replacements inside Real NVP and coupling-NSF architectures match or improve over the published affine and spline baselines on CIFAR10 and the UCI tabular suite, with the *spline+* hybrid (sinh conjugation followed by a rational-quadratic spline) often outperforming either bijection alone. Our experiments on $\phi^4$ lattice field theory (Section 5.5) demonstrate that coupling flows with our bijections scale to higher-dimensional problems with structure of scientific interest. On a $20 \times 20$ lattice, analytic bijections outperform affine baselines by around 10% in effective sample size. Moreover, problem-specific design—a $\mathbb{Z}_2$-symmetric zero-mode bijection trained separately—prevents mode collapse in the bimodal regime, illustrating how expressive bijections enable

---

[8]For coupling layers, the conditioner network must also be smooth to guarantee overall smoothness of the normalizing flow, and must not use activations such as ReLU.

architectural innovations tailored to the target distribution.

**Limitations and future work.** While our $\phi^4$ experiments validate coupling flows on lattice field theory, radial flows remain limited to low dimensions. We did not extensively tune hyperparameters for each method, focusing on fair comparison under fixed settings; each method could likely be improved with method-specific tuning. Several directions merit further investigation. First, hybrid architectures combining radial and coupling layers could leverage the strengths of both: radial layers for smooth, interpretable coarse structure and coupling layers for fine-grained corrections. Second, extending the Fourier parametrization to spherical harmonics would enable interpretable angular dependence in three dimensions, with potential applications to molecular and materials modeling, although combination with angle-transforming layers (Rezende et al., 2020) may be necessary. Finally, the analytic bijections, equivariant flows, and normalizing flows on manifolds are complementary rather than competing notions: the analytic bijections are scalar building blocks that may be used in equivariant or manifold architectures. Our $\mathbb{Z}_2$-symmetric zero-mode bijection and $O(n)$-equivariant angle-independent radial flow already illustrate this composition.

**Broader perspective.** The design space for normalizing flows is growing, with recent work exploring both strict constraints (exact analytic inverses) and relaxed constraints (e.g. free-form flows with numerical inversion (Draxler et al., 2024)). Our work demonstrates that principled construction of scalar bijections, guided by smoothness, invertibility, and expressivity requirements, yields practical benefits across multiple architectures. While we focused on coupling and radial flows, the bijections apply equally to autoregressive flows or other architectures requiring invertible scalar transformations. We hope these tools prove useful to practitioners seeking smooth, stable, and interpretable density estimation. Reference implementations of the analytic bijections and a tutorial notebook are available in the `bijx` normalizing-flow library (Gerdes & Cheng, 2025) at https://github.com/mathisgerdes/bijx.

## Impact Statement

This paper presents work whose goal is to advance the field of Machine Learning. There are many potential societal consequences of our work, none of which we feel must be specifically highlighted here.

## Acknowledgments

We thank Julian Ebelt and Kim Nicoli for helpful discussions. The research of MG and MC is supported by the Vici grant (number VI.C.232.117) from the Dutch Research

Council (NWO). MG was partially supported by the National Science Foundation under Cooperative Agreement PHY-2019786 (The NSF AI Institute for Artificial Intelligence and Fundamental Interactions, http://iaifi.org/).

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

# A. Detailed Derivations

This appendix provides detailed derivations for analytic inverses and Jacobians of the analytic bijections and radial flows introduced in the main text.

## A.1. Cubic Rational Bijection: Analytic Inverse

The cubic rational bijection is $y = h(x) = x + \lambda(x - \gamma)/(1 + (x - \gamma)^2/\sigma^2)$. Without loss of generality, set $\gamma = 0$ and define $\beta = 1/\sigma^2 > 0$, giving

$$y = x + \frac{\lambda x}{1 + \beta x^2}. \tag{16}$$

**Reduction to cubic equation.** Clearing denominators and rearranging yields

$$\beta x^3 - \beta y x^2 + (1 + \lambda)x - y = 0, \tag{17}$$

a general cubic with coefficients $a = \beta$, $b = -\beta y$, $c = 1 + \lambda$, $d = -y$.

**Cardano's formula.** For the general cubic $ax^3 + bx^2 + cx + d = 0$, the discriminants are $\Delta_0 = b^2 - 3ac$ and $\Delta_1 = 2b^3 - 9abc + 27a^2d$. For our coefficients,

$$\Delta_0 = \beta^2 y^2 - 3\beta(1 + \lambda), \qquad \Delta_1 = -2\beta^3 y^3 + 9\beta^2 y(1 + \lambda) - 27\beta^2 y. \tag{18}$$

The real root is

$$x = -\frac{1}{3a}\left(b + C + \frac{\Delta_0}{C}\right) = \frac{\beta y - C - \Delta_0/C}{3\beta}, \tag{19}$$

where $C = \sqrt[3]{(\Delta_1 \pm \sqrt{\Delta_1^2 - 4\Delta_0^3})/2}$ with sign chosen to maximize $|C|$. Our parameter constraints $-1 < \lambda < 8$ and $\beta > 0$ ensure $h$ is strictly increasing, guaranteeing a unique real solution.

**Derivation of parameter bounds.** Bijectivity requires $h'(x) = 1 + \lambda(1 - \beta x^2)/(1 + \beta x^2)^2 > 0$ for all $x$. Substituting $u = \beta x^2 \geq 0$ and defining $g(u) = (1 - u)/(1 + u)^2$, we find $g'(u) = (u - 3)/(1 + u)^3$ vanishes at $u = 3$ with $g(3) = -1/8$. Since $g(0) = 1$ and $g(u) \to 0^-$ as $u \to \infty$, we have $g(u) \in [-1/8, 1]$. Thus $h'(x) = 1 + \lambda g(u) > 0$ requires $1 - \lambda/8 > 0$ for $\lambda \geq 0$ and $1 + \lambda > 0$ for $\lambda < 0$, yielding $-1 < \lambda < 8$.

**Numerical stability.** The implementation avoids catastrophic cancellation by selecting the sign in $\Delta_1 \pm \sqrt{\Delta_1^2 - 4\Delta_0^3}$ that maximizes magnitude. The formula remains stable when $C \approx 0$ because this only occurs when $\Delta_0 \approx 0$ (nearly linear cubic).

## A.2. Scalar Bijection Derivatives

**Cubic rational.** For $h(x) = x + \lambda x/(1 + \beta x^2)$, the derivative is $h'(x) = 1 + \lambda(1 - \beta x^2)/(1 + \beta x^2)^2$. Under constraints $-1 < \lambda < 8$ and $\beta > 0$, we have $h'(x) > 0$ everywhere.

**Sinh bijection.** For $h(x) = \text{arcsinh}(e^\mu(e^\nu \sinh(x) + \delta))$, the log-Jacobian is

$$\log h'(x) = \mu + \nu + \log \cosh(x) - \frac{1}{2}\log(1 + \arg^2), \tag{20}$$

where $\arg = e^\mu(e^\nu \sinh(x) + \delta)$. We use $\log \cosh(x) = |x| + \log(1 + e^{-2|x|}) - \log 2$ for numerical stability.

**Cubic polynomial.** For $h(x) = g^{-1}(g(x) + \delta)$ with $g(x) = ax + bx^3$, the chain rule gives $h'(x) = g'(x)/g'(h(x)) = (a + 3bx^2)/(a + 3bh(x)^2)$. We compute this via automatic differentiation.

### A.3. Radial Flow Jacobian

**Direction-independent radial flow Jacobian.** For $g(\boldsymbol{x}) = f(r)\hat{\boldsymbol{x}}$ where $r = \|\boldsymbol{x}\|$ and $\hat{\boldsymbol{x}} = \boldsymbol{x}/r$, we compute $\partial g_i/\partial x_j$ using $\partial r/\partial x_j = \hat{x}_j$ and $\partial \hat{x}_i/\partial x_j = (\delta_{ij} - \hat{x}_i\hat{x}_j)/r$:

$$\frac{\partial g_i}{\partial x_j} = \frac{\partial}{\partial x_j}(f(r)\hat{x}_i) = f'(r)\hat{x}_j\hat{x}_i + f(r) \cdot \frac{\delta_{ij} - \hat{x}_i\hat{x}_j}{r} = \frac{f(r)}{r}\delta_{ij} + \left(f'(r) - \frac{f(r)}{r}\right)\hat{x}_i\hat{x}_j. \tag{21}$$

In matrix form, $J = \frac{f(r)}{r}I + \left(f'(r) - \frac{f(r)}{r}\right)\hat{\boldsymbol{x}}\hat{\boldsymbol{x}}^T$. Applying the matrix determinant lemma

$$\det(\alpha I + \beta \boldsymbol{u}\boldsymbol{u}^T) = \alpha^{n-1}(\alpha + \beta) \tag{22}$$

for $\|\boldsymbol{u}\| = 1$:

$$\log|\det J| = \log|f'(r)| + (n-1)\log\left|\frac{f(r)}{r}\right|. \tag{23}$$

**Angular-dependent radial flow Jacobian.** For $g(\boldsymbol{x}) = f(r, \hat{\boldsymbol{x}})\hat{\boldsymbol{x}}$, the log-Jacobian formula remains

$$\log|\det J| = \log\left|\frac{\partial f}{\partial r}\right| + (n-1)\log\left|\frac{f(r, \hat{\boldsymbol{x}})}{r}\right|. \tag{24}$$

In spherical coordinates $(r, \Omega)$, the transformation is $(r, \Omega) \mapsto (f(r, \Omega), \Omega)$ with $\det(J_{\text{sph}}) = \partial f/\partial r$. For the latter, note that the Jacobian in spherical coordinates is upper-triangular as $\mathrm{d}\Omega / \mathrm{d}r = 0$, so the determinant is the product of the diagonal terms and $\mathrm{d}\Omega / \mathrm{d}\Omega = 1$, leaving us with only the radial term. Via the chain rule, $J_{\text{Cart}} = J_{\text{sph}\rightarrow\text{Cart}}(f, \Omega) \cdot J_{\text{sph}} \cdot J_{\text{Cart}\rightarrow\text{sph}}(r, \Omega)$. The coordinate transformation determinants $\det(J_{\text{Cart}\rightarrow\text{sph}}) = 1/(r^{n-1}G(\Omega))$ and $\det(J_{\text{sph}\rightarrow\text{Cart}}) = f^{n-1}G(\Omega)$ contain a common angular factor $G(\Omega)$ that cancels, yielding

$$\det(J_{\text{Cart}}) = \left(\frac{f}{r}\right)^{n-1}\frac{\partial f}{\partial r}. \tag{25}$$

## B. Additional Numerical Results

### B.1. Bijection Visualizations

Figure 11 shows parameter sweeps for each bijection type (columns: $f(x)$, $|f'(x)|$, resulting density from standard normal). All analytic bijections can perform local deformations. In addition, sinh conjugation can create local stretches that propagate globally as a displacement of probability mass. Unlike affine transforms, these bijections can create localized peaks, asymmetric stretching, and multi-modal structures when composed. Table 4 summarizes the parameter notation across bijection types.

*Table 4.* Parameter notation across bijection types. All bijections share location $\gamma$ which controls the location where the transformation deviates non-linearly from the identity.

| Symbol | Role | Constraints | Used in |
|---|---|---|---|
| $\gamma$ | Location (center) | — | All |
| $\sigma$ | Scale (width) | $\sigma > 0$ | Rational, Sinh |
| $\lambda$ | Magnitude | $-1 < \lambda < 8$ | Rational |
| $\delta$ | Latent shift | — | Sinh, Cubic |
| $\mu, \nu$ | Global shift | — | Sinh |
| $a, b$ | Polynomial coefficients | $a, b > 0$ | Cubic |

### B.2. 1D Stack

Figure 12 shows the evolution of reverse KL divergence during training. All configurations exhibit stable training behavior. The benefit of additional bijections is clearly visible: a larger stack count leads to both faster initial convergence and lower final loss values. The training dynamics are similar across bijection types, suggesting that the performance differences in Figure 2 arise from representational capacity rather than optimization difficulty, although this should be further studied. All three bijection types are equally easy to train in this setting.

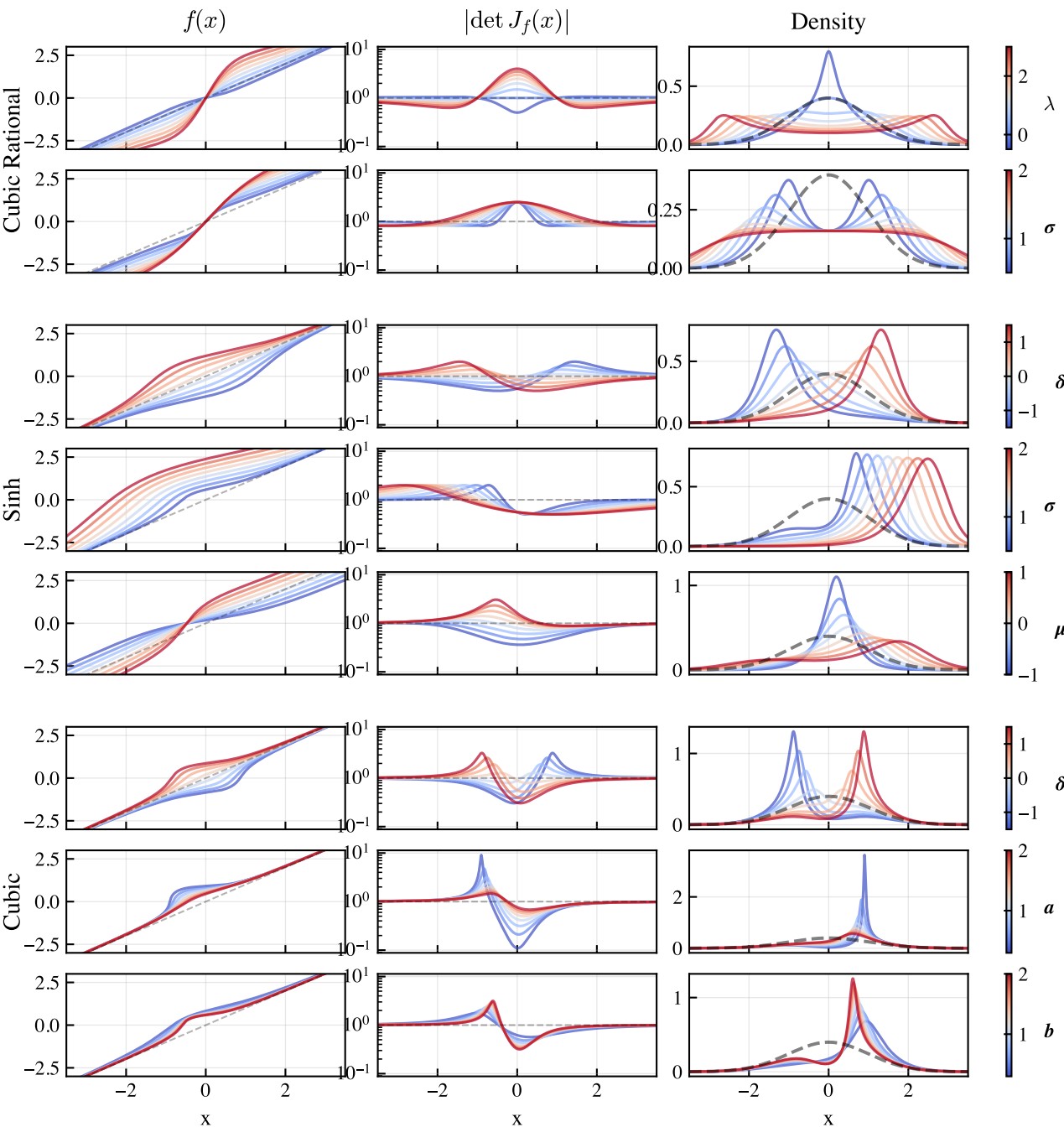

*Figure 11.* Parameter sweeps for the three analytic bijection types. Each row varies one parameter while holding others fixed. Left column: bijection function $f(x)$ (dashed line shows identity). Middle column: absolute Jacobian determinant $|f'(x)|$ on log scale (dashed line shows 1). Right column: resulting density when applied to a standard normal prior (dashed line shows prior). Color indicates parameter value from low (blue) to high (red).

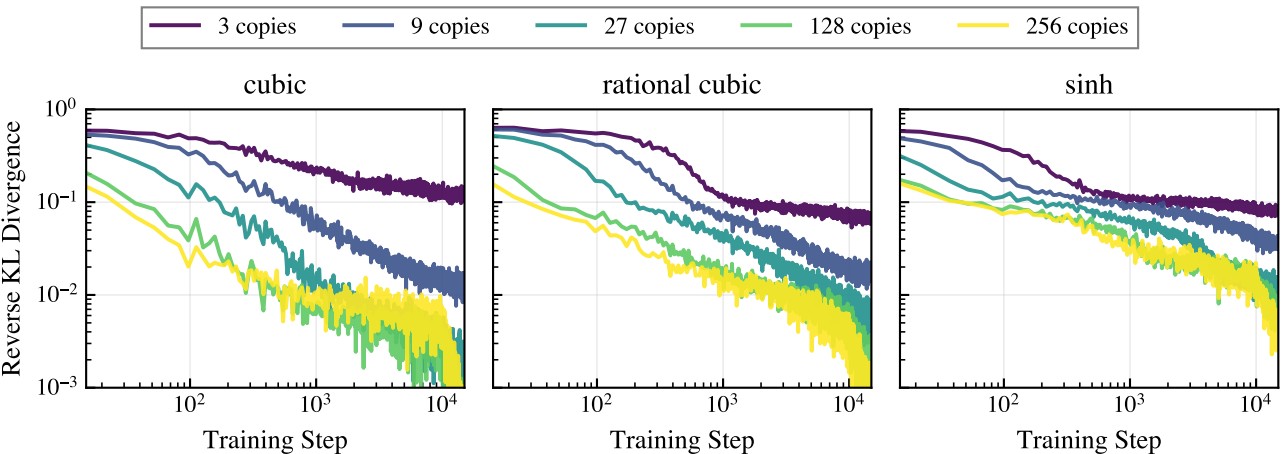

*Figure 12.* Training dynamics for 1D flows. Each panel shows reverse KL divergence vs. training steps for one bijection type, with different colors indicating the number of stacked bijections. All configurations converge stably, with more bijections generally achieving lower final loss. Each point shows the median of values pooled across 6 seeds within 15-step windows.

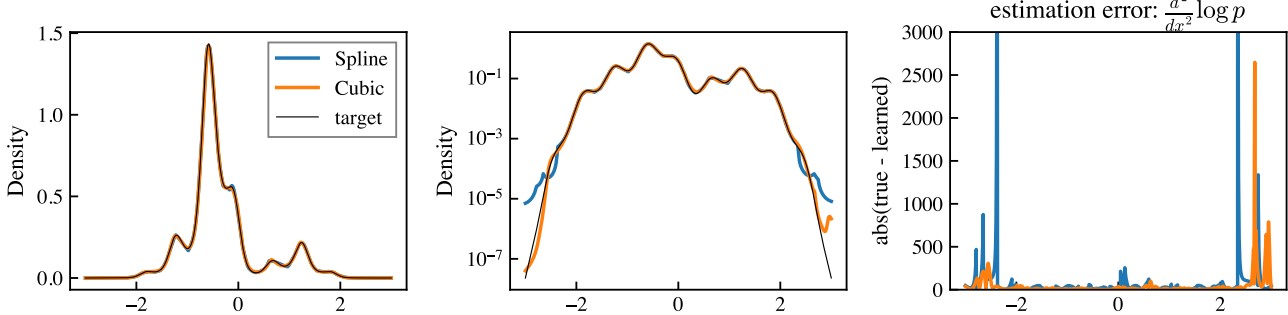

*Figure 13.* Density and second-derivative error for cubic conjugation (analytic) vs. rational-quadratic spline at matched ESS $> 99.9\%$ on a 1D oscillating target. *Left*: learned density. *Middle*: same in log-scale. *Right*: absolute error of $d^2/dx^2 \log p$. Both methods match the density well, but the spline shows large bin-boundary deviations in the second derivative, particularly near the tails.

### B.3. Smoothness and Higher-Order Derivatives

The desiderata for analytic bijections (Section 3) include global $C^\infty$ smoothness. Here we provide a concrete demonstration that this smoothness is observable in downstream quantities, even when two bijections fit a target density equally well. We train a stack of 42 cubic conjugation bijections and a stack of 3 monotonic rational-quadratic splines with 14 knots each on a 1D oscillating polynomial target. Both flows are trained with identical optimizer settings (Adam, exponential learning-rate decay from $5 \times 10^{-3}$ over 10,000 transition steps with decay rate 0.1, batch size 1024, 20,000 steps) and both reach final ESS above 99.9%.

**Second-derivative MSE.** Evaluating $d^2 \log p/dx^2$ on a uniform grid of 10,000 points over $x \in [-2.5, 2.5]$, the spline's MSE is $\approx 1445\times$ that of the cubic ($1/0.0006919 = 1445.30$). Reweighting under the target distribution itself (which de-weights the tails where the spline's bin-boundary errors are most extreme) the ratio drops to $\approx 8.1\times$. Either way, the spline's second derivative is markedly less faithful despite high fidelity density match. This matters in scientific applications that use gradient-derived quantities of $\log p$.

**Training-loss variance.** Figure 14 shows the forward KL and a shifted reverse-KL loss vs. training step (the reverse KL is shown on a log-scale, with an additive constant for visualization). While the forward KL curves are very similar between the two methods, the loss curve of the cubic conjugation displays a clear, monotonic reduction in noise as training progresses, whereas the spline's loss remains dominated by mini-batch noise of essentially constant magnitude. Quantitatively, the standard deviation of the loss over the last 5,000 steps is $1.5 \times 10^{-2}$ for the spline versus $7.3 \times 10^{-4}$ for the cubic conjugation,

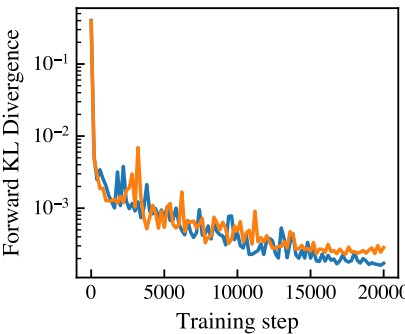 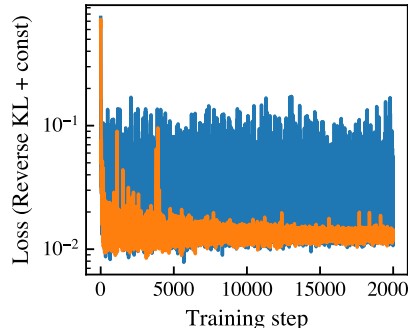

*Figure 14.* Training dynamics for cubic conjugation vs. rational-quadratic spline on the same 1D target. *Left*: forward KL vs. step. *Right*: loss (reverse KL up to an additive partition-function shift) in log-scale. The forward-KL curves are similar, while the cubic conjugation's loss variance decreases steadily over training, a feature absent in the spline variant.

about a $20\times$ reduction; this is consistent with the cubic's smoother optimization landscape.

### B.4. 2D Coupling Flows

Figure 15 shows training curves for 2D coupling flows. All configurations show stable convergence, with our bijections at stack count $N \geq 9$ achieving lower training loss than the affine and spline baselines.

### B.5. Coupling Layer Transformations

Figure 16 shows the progression through the 12 layers of trained coupling flows on the spiral target, comparing the Real NVP affine bijection (top) with a rational-quadratic spline bijection (bottom) substituted into the same coupling architecture. In both cases, the alternation between horizontal and vertical transformations is clearly visible: odd layers transform along one axis while even layers transform along the perpendicular axis, and the spiral structure emerges gradually through this sequence of axis-aligned operations. The spline-coupling progression still proceeds through alternating, axis-aligned coordinate updates, though its per-layer densities are visibly smoother than RealNVP's: the spline's locally curved bijection mitigates the most pronounced folding artifacts. The axis-aligned coupling structure itself, however, is shared by both architectures. In contrast, radial flows transform the distribution radially at each layer, more naturally matching the geometry of this target.

### B.6. Qualitative Analysis of Radial Flows

**Geometric constraints of radial flows.** A radial flow preserves rays: points along direction $\hat{x}$ from the center remain at that angle after transformation. This means the total probability mass in any angular wedge is conserved; a radial flow can redistribute mass *along* rays but not *between* them. For a single-center radial flow to approximate a target distribution, the target's angular distribution of mass (as viewed from the center) must approximately match the prior's.

For the spiral target used in Section 5.3, a single center suffices: the spiral arm winds around the origin, so every angular direction intersects the arm roughly equally. The total mass per angular wedge is approximately uniform, matching the isotropic Gaussian prior. The angular-dependent bijection then concentrates mass *onto* the spiral along each ray. In general, multiple layers of (angular-dependent) radial transformations may be needed. Stacking radial layers with different centers enables effective "angular mixing" through composition, as shown in Figure 8.

**Radius transformations.** Figure 17 visualizes how a trained radial Fourier flow transforms concentric circles centered at the origin. The inset shows the original circles in the base distribution; the main plot shows their images under the learned flow. The circles deform smoothly into the spiral structure of the target, with paths that never cross—a geometric consequence of the radial architecture. This non-crossing property ensures that nearby points in the base distribution remain nearby after transformation, contributing to the smoothness of learned densities.

Another way of visualizing the transformation is shown in Figure 18. Corresponding to the "Fourier expansion" of Figure 5, it shows how $f(r, \phi)$ varies across input radius and angle for each $K$, making explicit the effect of adding angular modes to

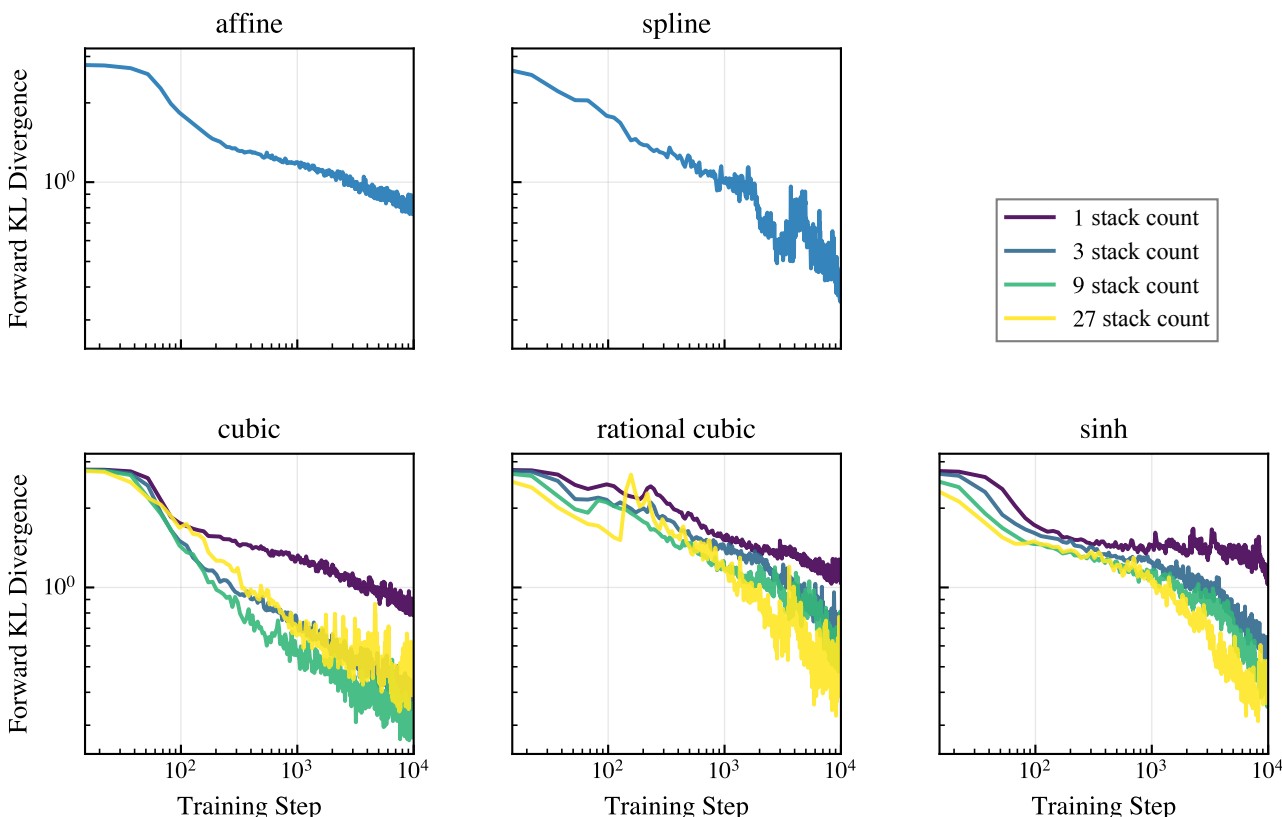

*Figure 15.* Training dynamics for 2D coupling flows. Each panel shows forward KL vs. training steps. Our methods (cubic, rational cubic, sinh) achieve lower or comparable training loss to affine and spline baselines with sufficiently many stacked bijections. Each point shows the median of values pooled across 6 seeds within 50-step windows.

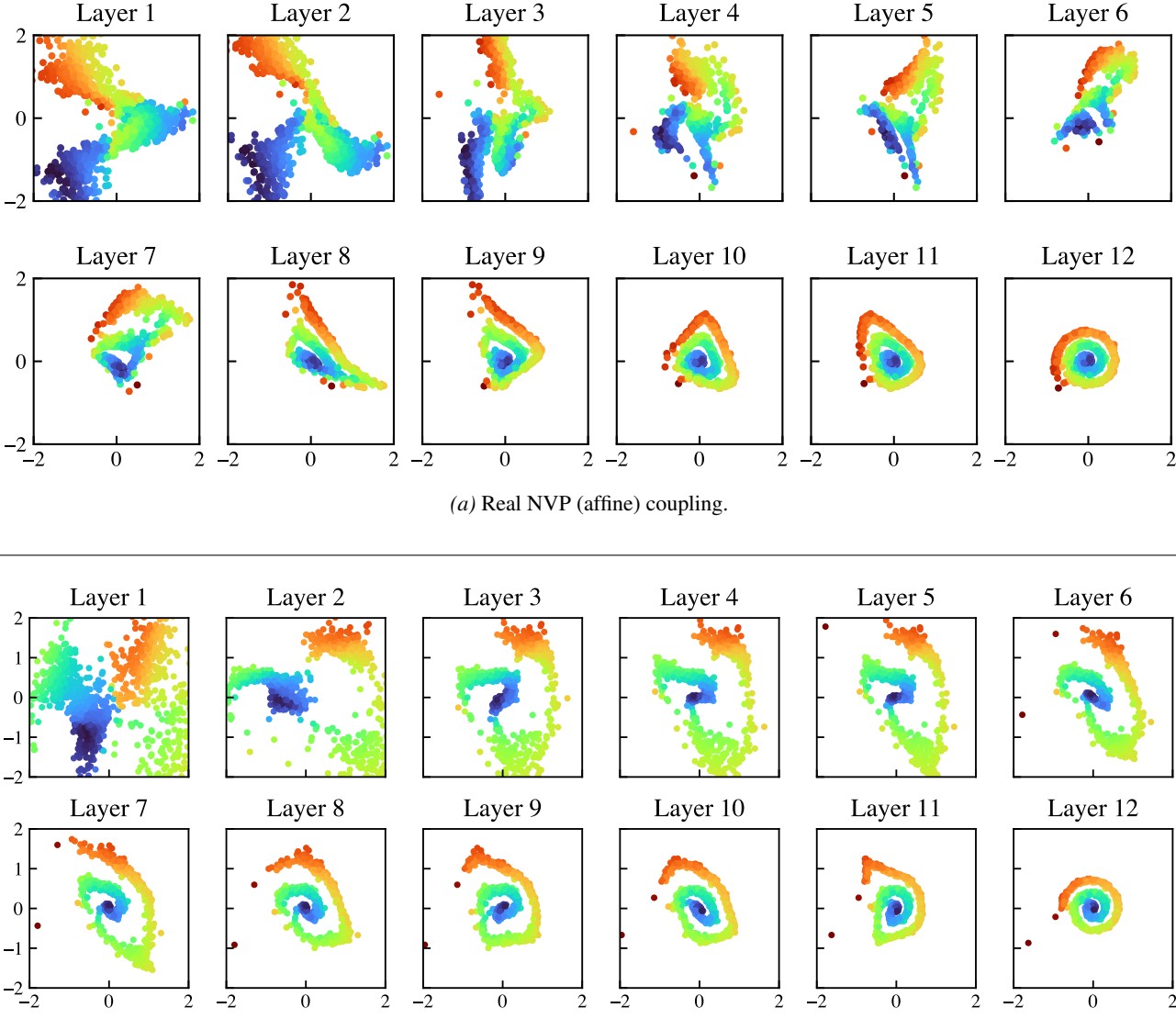

*(a)* Real NVP (affine) coupling.

*(b)* Rational-quadratic spline coupling in the same architecture.

*Figure 16.* Layer-by-layer progression through 12-layer coupling flows on the spiral target. Colors show final location in target distribution. Both architectures build the spiral through alternating axis-aligned operations: the affine variant shows the most pronounced folding, while the spline variant produces visibly smoother per-layer densities thanks to its more expressive non-linear bijection.

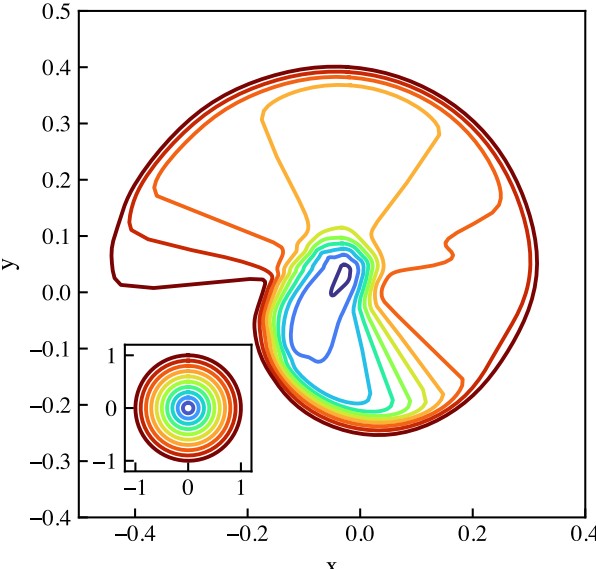

*Figure 17.* Transformation of points on a circle under a learned angular-dependent radial flow. Concentric circles (inset) transform smoothly into the spiral target structure. The color gradient indicates original radius, highlighting the preservation of radial ordering: paths never cross.

the flow.

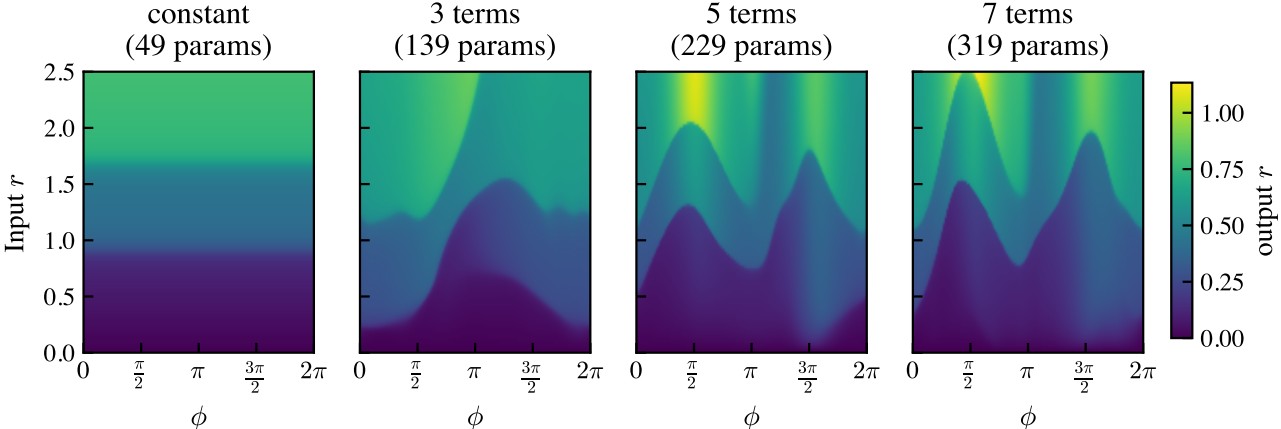

*Figure 18.* Radial transformation $f(r)$ as a function of input radius $r$ (y-axis) and angle $\phi$ (x-axis) for models with increasing Fourier terms. Color indicates output radius $r' = f(r, \phi)$. With one term (constant), the transformation is independent of $\phi$, yielding vertical color bands.

## B.7. Coupling Flows on $\phi^4$ Lattice Field Theory

Figure 19 supplements the final magnetization distributions of Figure 10 by showing training dynamics. The naive approach achieves high ESS ($\approx 90\%$) but this is misleading: all samples have negative magnetization, indicating complete mode collapse. This occurs because the reverse KL loss does not penalize missing modes, and the ESS (evaluated on samples from the model) only reflects the quality of samples within the covered mode.

In the pretrain approach, we first train the zero-mode bijection on $|\tilde{\phi}_0|$ with coupling layers frozen. This ensures 50% of samples have negative magnetization, as the $\mathbb{Z}_2$ symmetry is preserved by the zero-mode flow. After coupling layer training is turned on, the bimodal structure learned in pre-training is preserved.

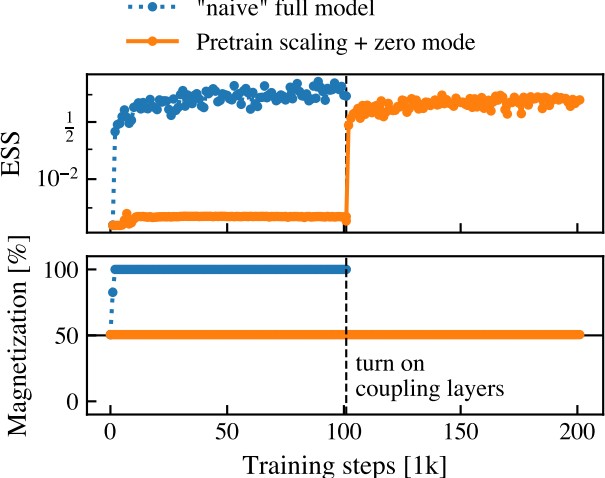

*Figure 19.* Training dynamics for bimodal $\phi^4$ distribution. **Top**: ESS (log scale). **Bottom**: Fraction of samples with negative magnetization. Naive training achieves high ESS but collapses to one mode (100% negative). Pretraining the zero-mode bijection maintains balanced sampling (50%) while achieving the same final ESS. Vertical dashed line marks when coupling layers are activated for the pretrain approach.

Figure 20 visualizes this collapse: samples from the naively trained network are uniformly negative (blue), while samples with zero-mode pre-training alternate between modes.

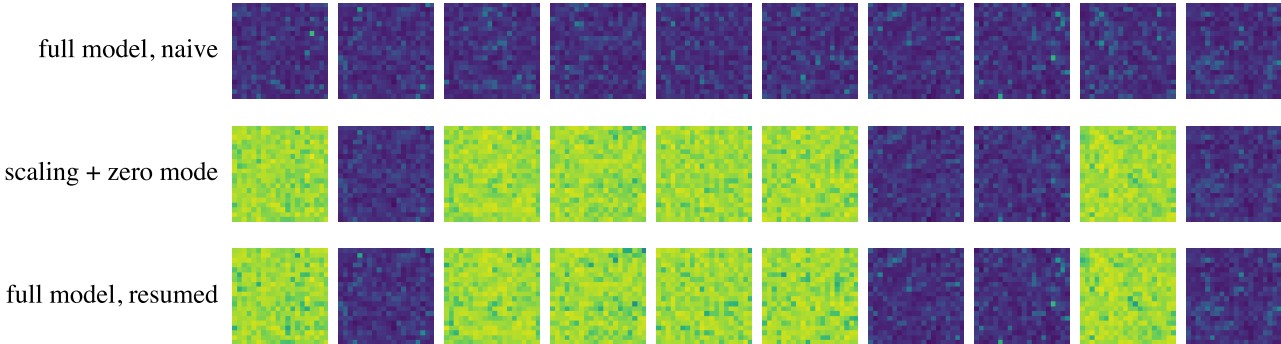

*Figure 20.* Lattice field configurations from trained models with 10 samples per row and same random seeds per column. **Top**: Naive training produces only negative-magnetization configurations (blue). **Middle**: After pretraining the zero-mode bijection, samples alternate between positive (yellow/green) and negative modes (blue). **Bottom**: Resumed training with coupling layers preserves bimodal sampling.

## B.8. Runtime and Scalability

We benchmark the wall-clock cost of each scalar bijection in isolation on GPU—no conditioner network, no gradient computation—so that the comparison reflects only the bijection itself. Figure 21 reports inference time as a function of the expressivity knob: number of knots for the spline, and stack count $N$ for our analytic bijections.

Memory footprint is essentially identical across all scalar bijections considered here: each transformation is element-wise and $O(1)$ in storage, so total memory in any coupling-flow application is dominated by the conditioner network, not by the bijection. At low stack counts the analytic bijections are cheaper than splines; the spline's roughly constant overhead is amortized once $N$ is large enough. Including gradient computation and JAX/XLA fusion in a full training step, the picture is more application-dependent: we observe end-to-end training cost ranging from comparable to spline (on $\phi^4$ with small MLP conditioners) up to roughly 1.5–2× spline (on CIFAR-10 with convolutional conditioners), in our unoptimized implementation.

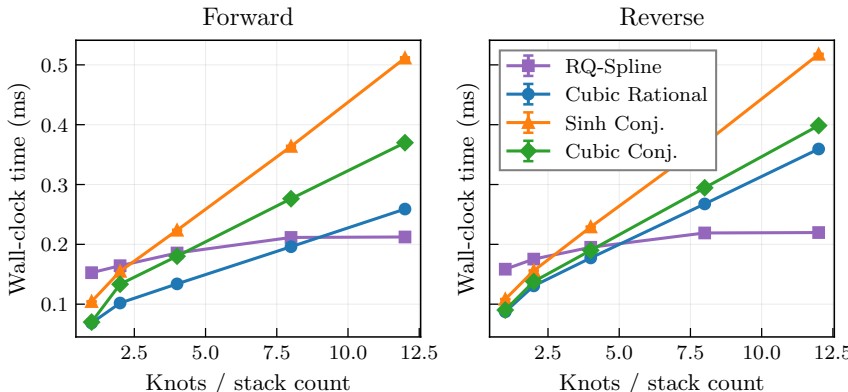

*Figure 21.* Inference cost of scalar bijections on GPU vs. expressivity knob. *Left*: forward direction. *Right*: inverse. Spline cost is essentially flat past $\sim 8$ knots; analytic bijection cost scales linearly with stack count $N$. Crossover points: cubic rational $\sim 8$ copies, cubic conjugation $\sim 4$, sinh $\sim 2$.

## C. Numerical Evaluation Details

### C.1. Metrics

We use three complementary metrics to assess flow quality:

- **Reverse KL divergence** $D_{\mathrm{KL}}(q\|p)$: Measures how well the flow $q$ covers the target $p$. Computed using the change-of-variables formula with Monte Carlo samples from the flow. Since samples are drawn from the flow itself, this loss is prone to mode collapse.

- **Forward KL divergence** $D_{\mathrm{KL}}(p\|q)$: Measures how well the flow matches the target, penalizing missing modes more heavily than reverse KL. For density estimation experiments (Section 5.2), this reduces to negative log-likelihood plus the (numerically estimated) target entropy.

- **Effective Sample Size (ESS)**: When the target density is known analytically, ESS quantifies how representative samples from the flow are for estimating expectations under the target. Defined as $\mathrm{ESS} = (\sum_i w_i)^2 / \sum_i w_i^2$ where $w_i = p(x_i)/q(x_i)$ are importance weights. If this is evaluated on samples from the trained model, as is typically done, note that this is not sensitive to mode missing (see also Section 5.5).

### C.2. Target Distributions

We use target distributions that highlight different aspects of flow expressivity:

- **1D oscillating polynomial**: A hand-crafted univariate density with multiple modes and varying scales, designed to test the ability of stacked bijections to capture complex structure.

- **2D spiral**: A continuous spiral distribution in two dimensions, providing a challenging test case that requires capturing curved, non-convex structure. This distribution has significant radial organization, making it suitable for both coupling and radial flow evaluation.

- **2D Gaussian mixture**: A mixture of 5 Gaussians arranged in a circular pattern, testing the ability to capture well-separated modes. This multimodal target complements the unimodal spiral.

- $\phi^4$ **lattice field theory**: A physics-motivated target on a $20 \times 20$ lattice with periodic boundary conditions, testing the scalability to higher dimensions and the ability to handle bimodal structure arising from spontaneous symmetry breaking.

**1D oscillating polynomial.** Unnormalized density:

$$\log p(x) = a_1 \sin(\omega_1 x)e^{-\gamma_1 x^2} + a_2 \cos(\omega_2 x) + a_4 x^4 \,, \tag{26}$$

with $a_1 = 1$, $a_2 = 2$, $a_4 = -0.2$, $\gamma_1 = 5$, $\omega_1 = 5$, $\omega_2 = 10$.

**2D spiral.** The density is defined implicitly through the sampling procedure:

$$x = t\cos(t) \cdot s + \epsilon_x, \quad y = t\sin(t) \cdot s + \epsilon_y, \tag{27}$$

where $t \sim \text{Uniform}(0, 5\pi)$, $s = 1/20$, $\epsilon_x, \epsilon_y \sim \mathcal{N}(0, 0.02^2)$. This produces a spiral with 2.5 turns.

**2D Gaussian mixture.**

$$p(x) = \frac{1}{K} \sum_{k=1}^{K} \mathcal{N}(x \,|\, \mu_k, \sigma^2 I), \tag{28}$$

with $\mu_k = R(\cos(2\pi k/K), \sin(2\pi k/K))^T$. We use $K = 5$, $R = 2.0$, $\sigma = 0.2$, yielding a pentagonal arrangement.

See Section 5.5 for the $\phi^4$ action.

### C.3. Numerical Implementation Details

The sinh bijection $h(x) = \sigma \cdot \text{arcsinh}(e^\mu(e^\nu \sinh((x - \gamma)/\sigma) + \delta)) + \gamma$ involves nested exponentials requiring careful numerical treatment.

**Asymptotic approximations.** For $|x| > T = 15$, we use asymptotic forms: $h(x) \approx x + \sigma(\mu + \nu)$ (positive $x$) or $h(x) \approx x - \sigma(\mu + \nu)$ (negative $x$), with $\log h'(x) \approx 0$.

**Stable special functions.** We compute $\log\cosh(x) = |x| + \log(1 + e^{-2|x|}) - \log 2$ to avoid overflow for $|x| \gtrsim 710$. For $\frac{1}{2}\log(1 + z^2)$, we use $\frac{1}{2}\texttt{log1p}(z^2)$ when $|z|^2 < 10^8$ and $\log|z|$ otherwise.

### C.4. Parametrization Details

We map unconstrained parameters $\theta_i \in \mathbb{R}$ to valid bijection parameters. In coupling layers, where a neural network generates the bijection parameters, controlling the initialization scale of the conditioner's final layer is critical. Standard initialization often yields parameter values that drive the bijection toward extreme transformations early in training. We mitigate this by initializing the flow close to the identity by scaling down the final layer weights or biases that feed into parameter transforms, and by compressing the parameter transforms, for instance using $\mu = \text{arcsinh}(\theta/10)$ instead of $\mu = \text{arcsinh}(\theta)$.

**Cubic rational** ($\epsilon_\alpha^{\text{low}} = \epsilon_\alpha^{\text{high}} = 10^{-3}$, $\epsilon_\beta = 10^{-1}$):

$$\gamma = \theta_0,$$
$$\lambda = \lambda_{\text{low}} + (\lambda_{\text{high}} - \lambda_{\text{low}}) \cdot \text{sigmoid}\left(\theta_1 + \text{logit}\left(\frac{-\lambda_{\text{low}}}{\lambda_{\text{high}} - \lambda_{\text{low}}}\right)\right), \tag{29}$$
$$\beta = \epsilon_\beta + \text{softplus}(\theta_2 + 1),$$

where $\lambda_{\text{low}} = -1 + \epsilon_\alpha^{\text{low}}$, $\lambda_{\text{high}} = 8 - \epsilon_\alpha^{\text{high}}$, and again $\beta = 1/\sigma^2$. The logit offset ensures $\theta_1 = 0 \Rightarrow \lambda = 0$.

**Sinh** ($\epsilon_\alpha = 0.1$):

$$\gamma = \theta_0, \quad \alpha = \text{softplus}(\theta_1) + \epsilon_\alpha, \quad \delta = \theta_2, \quad \mu = \text{arcsinh}(\theta_3), \quad \nu = \text{arcsinh}(\theta_4). \tag{30}$$

The arcsinh transform allows $\mu, \nu$ to take large values while dampening growth.

**Cubic polynomial** ($\epsilon_a = \epsilon_b = 10^{-2}$):

$$\gamma = \theta_0, \quad \delta = \theta_1, \quad a = \epsilon_a + \text{softplus}(\theta_2), \quad b = \epsilon_b + \text{softplus}(\theta_3). \tag{31}$$

**Design choices.** Epsilon values ($10^{-3}$ to $10^{-1}$) ensure strict positivity for numerical stability and well-defined gradients. For bounded intervals, sigmoid with affine rescaling and logit offset ensures initialization close to the identity.

*Table 5.* Training hyperparameters by experiment type.

| Experiment | Steps | Batch | Learning Rate | Schedule |
|------------|-------|-------|---------------|----------|
| 1D flows | 15,000 | 128 | $10^{-3}$ | Exponential decay |
| 2D coupling (spiral) | 5,000 | 256 | $4 \times 10^{-4}$ | Warmup (100 steps) + decay |
| 2D radial (spiral) | 10,000 | 128 | $5 \times 10^{-3}$ | Constant |
| Fourier radial (spiral) | 5,000 | 256 | $10^{-2}$ | Constant |
| 2D GMM (radial flows) | 5,000 | 256 | $10^{-2}$ | Constant |
| 2D GMM (coupling) | 5,000 | 256 | $5 \times 10^{-4}$ | Warmup (100 steps) + decay |
| $\phi^4$ coupling[†] | 100,000 | 64 | $10^{-3}$ | Constant |
| CIFAR10 (RealNVP+) | 100,000 | 64 | $10^{-4}$ | Constant |
| UCI tabular[‡] | 400,000 | 512 | $5 \times 10^{-4}$ | Cosine decay |

[†]Pretrain phase trains only FFT scaling and zero-mode bijection; coupling layers are frozen. [‡]UCI defaults shown; MINIBOONE uses 200,000 steps, batch 128, learning rate $3 \times 10^{-4}$.

## C.5. Optimization Details

Table 5 summarizes the training hyperparameters used across experiments. All experiments use the Adam optimizer with default momentum parameters ($\beta_1 = 0.9$, $\beta_2 = 0.999$).

Exponential decay schedules decay learning rate by factor 10 over training; warmup linearly increases from 0 over 100 steps. The higher learning rates for radial flows ($5 \times 10^{-3}$ to $10^{-2}$) vs. coupling flows ($4 \times 10^{-4}$) reflect superior training stability.

## C.6. Architecture Overview

2D coupling: 12 layers, alternating masks, 2-layer dense ResNet conditioners (128 hidden units, GELU). All bijection types use identical conditioners for fair comparison. They only differ in their final output dimensions to match the number of parameters for each chosen bijection type. Radial flows: learned centers from $\mathcal{N}(0, 1)$, stacked scalar bijections; near-zero initialized coefficients (bijections near identity). GMM comparison: coupling (16 layers, 12 bijections/layer, 2311k params), angular radial (32 centers, 5 Fourier terms, 12 bijections/center, 7.8k params), pure radial (32 centers, 12 bijections/center, 1.6k params). $\phi^4$: 12 coupling layers with checkerboard masks, small ConvNets (2 layers, 16 channels, $3 \times 3$ kernels, circular padding), FFT per-shell scaling; "realNVP+" uses 8 analytic + 1 affine bijection per layer; bimodal adds $\mathbb{Z}_2$-symmetric zero-mode bijection (8 cubic bijections on $k = 0$).

**CIFAR10 (RealNVP+).** A multi-scale architecture broadly following Real NVP (Dinh et al., 2017): three scales, each consisting of a squeeze operation followed by an ActNorm layer, a stack of channel-split coupling layers, and a factor-out step. The conditioner is a convolutional ResNet (8 residual blocks, 64 hidden channels, $3 \times 3$ kernels, GELU activations) initialized near the identity. Inputs are mapped to the unbounded domain by a logit preprocessing with $\alpha = 0.05$. Each coupling layer stacks 8 analytic scalar bijections (cubic rational, cubic conjugation, or sinh conjugation) followed by a single affine transformation, which we refer to as "RealNVP+". Deviations from the original Real NVP architecture include using ActNorm rather than batch normalization, GELU rather than ReLU activations, the absence of a learned prior, and the absence of weight normalization.

**UCI tabular.** The coupling architecture follows the RQ-NSF(C) configuration of Durkan et al. (2019), with per-dataset settings: POWER and GAS use 10 coupling layers and a ResNet conditioner of width 256 with two residual blocks; HEPMASS and BSDS300 use 20 coupling layers and width 128 with one residual block; MINIBOONE uses 10 coupling layers, width 32, one residual block, and 4 spline bins (vs. 8 elsewhere). Per-dataset dropout matches Durkan et al. (2019). We train three variants in this architecture: *spline* uses a single rational-quadratic spline; *sinh* stacks sinh conjugation bijections followed by a trailing affine; *spline+* stacks sinh conjugation bijections followed by a rational-quadratic spline. The number of stacked sinh bijections is held fixed per dataset across the three variants but varies slightly between datasets (reduced for GAS and MINIBOONE, where added depth tended to destabilize training). Our reimplementation in JAX/`nnx` differs in minor ways from Durkan et al. (2019) (primarily in initialization) which affects the reproduced *spline* baseline equally across variants.

