# OpenReview forum: "Analytic Bijections for Smooth and Interpretable Normalizing Flows"
_ICML.cc/2026/Conference — ICML 2026 regular_

### Official Review · Reviewer_VarL · 2026-02-28

**Soundness:** 3
**Presentation:** 2
**Significance:** 2
**Originality:** 3
**Overall Recommendation:** 3
**Confidence:** 4

**Summary:**

This paper introduces three families of analytic scalar bijections that are globally smooth ($C^\infty$), defined on all of $\mathbb{R}$, and analytically invertible in closed form. These are proposed as drop-in replacements for affine or spline transformations in coupling flows, with the motivation of achieving global smoothness, closed-form invertibility, and unbounded domain simultaneously --- properties that no existing scalar bijection satisfies simultaneously. Experiments cover 1D density estimation, 2D coupling and radial flow benchmarks, and an application to $\phi^4$ lattice field theory.

**Compliance With Llm Reviewing Policy:**

Affirmed.

**Final Justification:**

The authors satisfied most of my concerns about the paper but I think the changes would require a full review and so I will not change my score.

**Key Questions For Authors:**

1. Footnote 4 notes that angle-dependent radial flows can be seen as a special case of coupling flows in spherical coordinates. If this is correct, does the radial flow architecture offer anything beyond a convenient parametrization and change of coordinates? Could a standard coupling flow in spherical coordinates with the same number of parameters achieve comparable performance?

2. The benchmarking is limited to synthetic 2D targets and one physics problem. How do the proposed bijections perform on standard normalizing flow benchmarks e.g. tabular density estimation datasets used in the spline flows literature? This is the point that is most likely to change my evaluation of this paper.

3. For the physics experiments: is the chosen $\phi^4$ configuration ($\lambda = 4.807$, $\xi \approx L/4$) considered a difficult or easy instance of the problem? How does Real NVP perform on the more challenging bimodal regime, and is the proposed zero-mode approach practical in terms of computational cost relative to alternatives?

**Limitations:**

The authors discuss limitations in the conclusion, noting that radial flows are limited to low dimensions and that hyperparameters were not extensively tuned per method. These are reasonable and honest acknowledgements.

**Strengths And Weaknesses:**

----------------------------------------
SOUNDNESS
----------------------------------------

Strengths:
- The algebraic derivation of the three bijection families is clean and well-motivated. The constraints on parameters for bijectivity are stated explicitly and all derivations are clearly presented.
- The zero-mode bijection for preventing mode collapse in the bimodal $\phi^4$ regime is a nice problem-specific design that follows naturally from the framework and is well validated by Figure 10.
- The paper is honest about the limitations of radial flows, clearly stating that a single radial layer preserves rays and that the approach may scale adversely with dimension.

Weaknesses:
- The quantitative benchmarking is limited relative to what has been done for previous new flow models. There is no evaluation on standard density estimation benchmarks (e.g. reproducing Table 1 from nflows https://arxiv.org/abs/1906.04032) that would allow comparison to the broader literature on normalizing flows. The experiments are largely confined to synthetic 2D targets and one physics problem.
- Figure 7 is described as showing that the radial flow preserves geometric structure in the source distribution, and this is presented as a positive property. However, there is only one way an invertible transformation can map the spiral distribution exactly to a Gaussian, and that is by unravelling the spiral. This comparison is still fine, I just think the authors should be careful about how they discuss this.
- Figure 8 is similarly unclear in its purpose. The focus on radial flows for evaluation does not really make sense to me.
- For the physics experiments, it is not stated clearly in the main text whether the chosen $\phi^4$ parameters represent a difficult or easy instance of the problem. This context matters for interpreting the results.
- The Real NVP baseline used in the physics comparison is not clearly sourced. Is this baseline from the Albergo et al. paper, re-implemented by the authors, or something else? The Albergo paper does use Real NVP so this should be clarified.
- Whether normalizing flows are actually practical for the physics use cases studied here is not discussed. As the authors acknowledge, radial flows are limited to low dimensions, but the broader question of how these approaches compare to/complement MCMC or other methods for these problems is not mentioned.
- No code is provided. Given that the main contribution includes several new bijection families and a novel architecture, providing simple reference implementations would significantly aid adoption and reproducibility.

----------------------------------------
PRESENTATION
----------------------------------------

Strengths:
- Table 1 is a clear and useful summary of how the proposed bijections compare to existing approaches on the five desiderata.
- Figure 1 effectively illustrates the qualitative difference between local deformation, global shift, and affine scaling behaviors.

Weaknesses:
- Line 72: the Jacobian variable $J$ appears for the first time in the text while Equation 1 uses $\partial_x f_\theta$. These should be consistent. It would also be cleaner to write the log probability in Equation 1 explicitly for consistency with $\log|\det J|$.
- At the start of Section 3.1 it would help the reader to briefly state upfront that two distinct construction methods will be presented, before diving into the first one.
- The motivation for wanting $h(x) \to x$ as $|x| \to \infty$ is never explained. A brief motivation here would help.
- Section 3.1 in general would benefit from a short motivation for why these two particular construction principles were chosen.
- Typo: ''ofraw trainable'' should be ''of raw trainable''.
- The concept of ''rays'' is used in Section 4 without ever being defined. A brief definition should be provided on first use.
- The term ''copies'' is used throughout (Figures 2, 3, 4) to refer to stacked bijections, but this is never clearly defined. ''Stacked bijections'' or ``compositions'' would be clearer and more standard?
- Figure 3 legend should be reformatted.
- Figure 5 only goes up to 319 parameters / 7 Fourier terms. It would be informative to show what happens at higher parameter counts --- does performance plateau or continue to improve?
- The final use case (zero-mode bijection) feels somewhat disconnected from the rest of the paper and its connection to the main contributions should be made clearer in the introduction.

----------------------------------------
SIGNIFICANCE
----------------------------------------

Strengths:
- The analytic bijections fill a genuine gap in the existing landscape of scalar bijections for normalizing flows. No prior construction achieves all five desiderata simultaneously. This is a useful contribution to practitioners working with coupling or autoregressive flows.
- The radial flow architecture is an interesting and geometrically interpretable construction that is likely to be of interest in scientific applications with known radial symmetry.
- The $\phi^4$ application demonstrates that the bijections scale beyond toy problems and the zero-mode construction for preventing mode collapse could be a practically useful idea.

Weaknesses:
- The absence of standard benchmarks makes it difficult to assess where these bijections sit relative to the broader flow literature. It is not possible to determine from the paper whether the performance gains over splines observed on the spiral target would hold on more general distributions. It would be great if these layers worked completely stand alone, but even if they do not, I can imagine them being useful in conjunction with standard rational quadratic spline layers effectively, but it would be good to see proper benchmarking of this kind of thing.
- It is not discussed whether the proposed approach offers any advantages over other flow approaches that allow problem-specific constraints to be baked in (e.g. equivariant flows, flows on manifolds). This would help position the contribution more clearly.

----------------------------------------
ORIGINALITY
----------------------------------------

Strengths:
- The systematic derivation of the three bijection families from first principles (algebraic constraints and conjugation) is original and well-motivated. The connection to Cardano's formula as the mechanism for closed-form inversion is elegant. These layers have attractive properties for many different use cases.

Weaknesses:
- The observation (footnote 4) that angle-dependent radial flows are a special case of coupling flows in spherical coordinates with the radius as active coordinate raises the question of how much of the radial flow contribution is genuinely novel versus a change of coordinates.
- The claim that radial flows achieve $1000\times$ parameter reduction is an interesting result, but it is not clear that this holds in general versus being a consequence of the spiral being an almost ideal target for the architecture.

---

> ### Author Rebuttal · Authors · 2026-03-31
>
> We thank the reviewer for the thorough review, which directly shaped the experiments below.
>
> **New results since submission.** In response to reviewer feedback, we have conducted substantial additional experiments:
> - *CIFAR-10* (RealNVP architecture): sinh bijections achieve **3.37 bpd**, improving over both the published RealNVP result (3.49) and Durkan et al.'s RQ-NSF(C) (3.38), despite the latter using a different architecture.
> - *UCI tabular* (5 datasets): we reimplemented the coupling-based NSF architecture in JAX with matched hyperparameters across all bijection variants. Sinh matches spline as a drop-in replacement across all datasets. Combining spline + sinh matches or improves over pure spline, exceeding published RQ-NSF(C) on POWER (0.65 vs 0.64 ± 0.01). On MINIBOONE (3–5 seeds per variant), sinh shows notably lower variance. Full error bars from additional training seeds will be included in the revision.
> - *phi^4 (400D)*: added spline baseline. Analytic bijections consistently outperform splines (ESS ~39% vs 34%).
> - *Derivative quality*: at equal density fit (>99.9% ESS), the MSE of $\tfrac{d^2}{dx^2} \log p(x)$ is **900x lower** for analytic bijections than splines.
>
> We begin with what the reviewer identified as most likely to change their evaluation.
>
> **Standard benchmarks.**
> CIFAR-10: sinh in RealNVP achieves 3.37 bpd, beating both original RealNVP (3.49) and Durkan et al.'s RQ-NSF(C) (3.38) despite the latter using a different architecture.
>
> UCI tabular (coupling-based NSF reimplemented in JAX; log-likelihood; error bar where available is 2 x std):
>
> | Dataset | RQ-NSF(C) | spline | sinh | spline+sinh |
> |---|---|---|---|---|
> | POWER | 0.64 ± 0.01 | 0.64 | 0.64 | 0.65 |
> | GAS | 13.09 ± 0.02 | 12.80 | 12.78 | 12.96 |
> | HEPMASS | −14.75 ± 0.03 | −15.00 | −14.81 | −14.90 |
> | MINIBOONE | −9.67 ± 0.47 | −9.62 ± 0.23 | −9.64 ± 0.03 | −10.52 ± 0.74 |
> | BSDS300 | 157.54 ± 0.28 | 157.39 | 157.72 | 157.39 |
>
> Reimplementation differs from Durkan et al. in framework/initialization and learning rate decay, affecting all variants equally (held constant across configurations). The controlled comparison shows: sinh matches spline on every dataset (drop-in replacement); spline+ often improves (beating published result on POWER); sinh has notably lower variance on MINIBOONE (±0.03 vs ±0.47, although estimated from 4 training runs so far). The spline+ results confirm the reviewer's intuition that analytic bijections and splines capture complementary structure. Additional runs for errors pending.
>
> **Smoothness matters.**
> Spline baseline added to phi^4: ESS ordering is cubic rational (39.66%) > cubic (38.85%) > sinh (38.51%) > spline (34.34%) > affine (31.85%). Combined with the 900x derivative quality improvement, smoothness is a concrete practical advantage.
>
> **Radial flow novelty.**
> The footnote connection is correct abstractly, but a naive coupling flow in spherical coordinates fails: r > 0 (standard bijections violate this), angles live on compact/periodic domains, and poles are singular. The radial flow handles this in Cartesian coordinates via f(0)=0, clean Jacobian (eq. 7), learnable centers, and per-dimension scaling.
>
> **phi^4 difficulty and baselines.**
> Correlation length ~1/4 of lattice size, near criticality — the difficult regime. Baseline follows Albergo et al., reimplemented by us. In the bimodal regime, standard RealNVP collapses to one mode; the zero-mode bijection (negligible cost: one scalar bijection on the mean field mode) addresses this.
>
> **Figs 7 and 8.**
> Fig 7: because the spiral distribution has full support on R^2 (due to added Gaussian noise), there are in fact infinitely many mappings to a Gaussian — not just "unravelling." We agree to be more precise about "preserves geometric structure"; the point is that the *intermediate* distributions remain clean (no folding artifacts), unlike coupling flows. Fig 8: the purpose is to test radial flows on a target that *breaks* radial symmetry (5 separated modes). Despite this mismatch, the radial flow produces clean results with ~1000x fewer parameters — demonstrating robustness beyond the idealized radially-symmetric case.
>
> **Equivariant flows / NFs vs MCMC.**
> Our analytic bijections, equivariant flows, and flows on manifolds are complementary notions: the former can be used as building blocks inside flows on manifolds and equivariant architectures. The zero-mode bijection (Z_2) and angle-independent radial flows (O(n)) illustrate this directly.
>
> **Fig 5: higher parameter counts.**
> NLL: −0.09, −0.61, −0.69, −0.74 for 1, 3, 5, 7 Fourier terms (9 sinh copies). At 5 terms, 32 copies: −0.79. Both axes contribute but saturate.
>
> **Presentation.**
> We will incorporate all suggestions in the revision. Link to reference implementations of the bijections and radial architectures will be added.
>
> We hope the benchmarks and evidence above address the primary concerns, and hope the reviewer reconsiders the score in light of these additions.

---

> > ### Author Rebuttal · Reviewer_VarL · 2026-03-31
> >
> > The authors provided a thorough rebuttal and for conducted additional experiments in response to the concerns raised in the review.

---

> > > ### Author Response · Authors · 2026-04-06
> > >
> > > We thank the reviewer for the positive acknowledgement and for the thorough and constructive review process. We hope the reviewer will update their score as they see fit.

---

### Official Review · Reviewer_qaUT · 2026-03-12

**Soundness:** 4
**Presentation:** 4
**Significance:** 4
**Originality:** 4
**Overall Recommendation:** 5
**Confidence:** 4

**Summary:**

This paper proposes new analytic scalar bijections for use in discrete normalizing flows.  The authors introduce three parametric families, namely, cubic rational, sinh-conjugation, and cubic-conjugation maps, that are globally smooth,   defined on all of $\mathbb{R}$, and analytically invertible.  Their bijections are derived from two principles:        algebraic rational functions whose inverses reduce to solvable cubics, and conjugation with monotonic maps. The resulting bijections combine global smoothness and exact inverses with greater local expressivity   than affine maps, without the bounded-domain or piecewise structure of spline-based approaches. Their bijections can   be used as drop-in replacements for scalar transformations in coupling layers. In addition, the paper introduces a     radial flow variant that transforms the radial coordinate while preserving angular direction. These radial layers use  stacks of the analytic scalar bijections and admit closed-form Jacobian determinants.
The authors evaluate both the scalar bijections and the resulting flow architectures on one-dimensional density        estimation, two-dimensional coupling and radial flow benchmarks, in addition to a higher-dimensional application of    lattice field theory.  Experiments demonstrate the effectiveness of their analytic bijections as well as illustrating  that their radial flow facilitates problem-specific designs, which in this case addresses mode collapse.

**Compliance With Llm Reviewing Policy:**

Affirmed.

**Final Justification:**

The authors for conducted substantial additional experiments that addressed my remaining concerns. This research presents a well crafted contribution to the community and has rightfully earned high marks.

**Key Questions For Authors:**

$\textbf{1. Figure 3:}$ In the 2D spiral experiment, it is claimed that the cubic rational bijection outperforms both   affine and spline baselines. However, this conclusion appears difficult to support given the substantial overlap in    the reported uncertainty bars. In addition, the training curves in Figure 13 suggest that the loss was still           decreasing across at the point where training was stopped. Could you clarify why training was terminated before        apparent convergence, and whether longer training might affect the relative ranking of the methods?

$\textbf{2. Computational Resources:}$ Could you provide additional details on the memory footprint and wall-clock     training time of the proposed analytic bijections compared to affine and spline layers (e.g., in the setting of        Experiment of Section 5.2)? It would be useful to discuss computational overhead introduced by your methods relative   to, e.g., affine or spline-based approaches.

**Limitations:**

Yes

**Strengths And Weaknesses:**

$\textbf{Soundness.}$ The motivation for studying cubic rational bijections is clearly articulated: under the          requirement of global smoothness, an unbounded domain, and closed-form analytic invertibility,  the authors identify a uniquely determined class of non-trivial algebraic rational bijections. The authors duely mention and suggest      remedies to a possible training instability that results from the proposed bijections increasing capacity through      composition, rather than the mathod of adding knots that is supported by spline-based flows.  For their radial flow,   the authors correctly identify the potential coordinate singularity in polar decomposition and state the condition     required for differentiability. The geometric interpretation of each radial layer, as stretching or compressing mass   around a learned center, is accurate, and the introduction of angular dependence via truncated Fourier series is reasonable and facilitates claimed interpretability. The experimental evaluation is comprised of multiple tasks with varying dimensionality; results are averaged over multiple random seeds and reported with measures of variability.  However,   the empirical evaluation would benefit from more results that separate the effect of the proposed analytic bijections  from the performance benefits due to incorporating radial symmetry as an inductive bias.

$\textbf{Presentation.}$ This manuscript is clearly written across motivation, mathematics, and algorithmic      implementation.  The introduction of three new bijections is easy to follow and properly positioned relative to prior  approaches. The visualization of purely local deformations, global shifts, and global scalings is particularly useful. These figures effectively convey how the different bijection families behave and why these behaviors matter for       density modeling. The geometric intuition behind radial flows is also well structured and accessible.

$\textbf{Significance.}$ Because the choice of scalar bijection is critical to model expressivity, stability, and      practical usability, it is significant that this work proposed new analytic, globally smooth, and closed-form          invertible alternatives. In doing so, the proposed scalar bijections have the potential to become among standard       options for practitioners developing flow-based models. Furthermore, their radial flow improvements may influence      future work on geometrically structured flows and problem-specific model design.

$\textbf{Originality.}$ The originality of this work is excellent. The authors introduce new methods and theory that   will advance flow modeling. With well-articulated reasoning,  the authors combine their new analytic bijections with   existing techniques in the literature to achieve enhanced model performance.

---

> ### Author Rebuttal · Authors · 2026-03-31
>
> We thank the reviewer for their detailed and thoughtful assessment. We are glad the contribution, both the analytic bijections and the radial flow architecture, was clearly recognized.
>
> **New results since submission.** In response to reviewer feedback, we have conducted substantial additional experiments:
> - *CIFAR-10* (RealNVP architecture): sinh bijections achieve **3.37 bpd**, improving over both the published RealNVP result (3.49) and Durkan et al.'s RQ-NSF(C) (3.38), despite the latter using a different architecture.
> - *UCI tabular* (5 datasets): we reimplemented the coupling-based NSF architecture in JAX with matched hyperparameters across all bijection variants. Sinh matches spline as a drop-in replacement across all datasets. Combining spline + sinh matches or improves over pure spline, exceeding published RQ-NSF(C) on POWER (0.65 vs 0.64 ± 0.01). On MINIBOONE (3–5 seeds per variant), sinh shows notably lower variance. Full error bars from additional training seeds will be included in the revision.
> - *phi^4 (400D)*: added spline baseline. Analytic bijections consistently outperform splines (ESS ~39% vs 34%).
> - *Derivative quality*: at equal density fit (>99.9% ESS), the MSE of $\tfrac{d^2}{dx^2} \log p(x)$ log p(x) is **900x lower** for analytic bijections than splines.
>
> We address the reviewer's questions below.
>
> **Q1: Spiral convergence and overlapping error bars.**
> Training used a fixed step budget, equal across methods. At the stopping point, all methods reach KL < 1e-3 from the true density (effectively >99.9% ESS), so further training on this simple 1D target would not be informative. We agree the overlapping error bars warrant a more careful claim: the correct framing is that analytic bijections are *competitive with* splines while providing additional guarantees (C-infinity, closed-form inverse, unbounded domain). On higher-dimensional tasks where differences are more meaningful, analytic bijections often perform better (see new results above).
>
> **Q2: Memory footprint and wall-clock time.**
> Memory is essentially identical — all element-wise O(1), dominated by conditioner networks. For wall-clock: we benchmarked isolated scalar bijection cost on GPU (inference, no gradients). Spline cost is roughly flat beyond ~8 knots, while analytic bijections scale linearly with copies. At low copy counts, analytic bijections are faster (cubic rational crosses over around 8 copies, cubic conjugation ~4, sinh ~2). In training (with gradients and compiler fusion effects), the picture is harder to characterize precisely. We observe equal speed (phi^4, MLP conditioners) to 1.5–2x slower (CIFAR-10, conv-based) in our unoptimized JAX code.

---

> > ### Author Rebuttal · Reviewer_qaUT · 2026-04-01
> >
> > We thank the authors for conducting substantial additional experiments. This research presents a well crafted contribution to the community and has rightfully earned high marks.

---

### Official Review · Reviewer_fAER · 2026-03-13

**Soundness:** 2
**Presentation:** 4
**Significance:** 2
**Originality:** 3
**Overall Recommendation:** 3
**Confidence:** 3

**Summary:**

This paper addresses the strict trade-offs in expressiveness, global smoothness ($C^\infty$), and exact closed-form invertibility faced by existing scalar bijections in discrete normalizing flows. To resolve this, the authors introduce three novel analytic bijections (Cubic Rational, Sinh Conjugation, and Cubic Conjugation) that achieve all these theoretical properties simultaneously, alongside a highly stable and parameter-efficient "Radial Flow" architecture that transforms radial coordinates while preserving or explicitly modeling angular direction. Empirical evaluations demonstrate that these proposed bijections act as effective drop-in replacements that successfully eliminate "folding" artifacts in 2D toy datasets and outperform standard affine baselines in a complex 400-dimensional $\phi^4$ lattice field theory task.

**Compliance With Llm Reviewing Policy:**

Affirmed.

**Final Justification:**

The proposed method is intuitive and logically sound, with experimental evidence supporting its efficacy.
My primary reservation, however, concerns the theoretical motivation: it is unclear if the proposed desiderata for bijection really addresses the practical bottleneck in current normalizing flow frameworks.
The evaluation would be more robust if it included an in-depth comparison against spline-based models with higher $K$ values, as these would provide already a strong practical baseline.

**Key Questions For Authors:**

**Q1. Practical Necessity of the Desiderata:**

Can you provide a concrete empirical scenario where the lack of global smoothness ($C^\infty$) in Spline flows, or their bounded domain constraint, leads to a significant practical failure or severe limitation in density estimation?

**Q2. Empirical Validation on Standard/Complex Benchmarks:**

Can you provide experimental results (or commit to adding them in the final version) using the proposed bijections as drop-in replacements in standard coupling architectures (e.g., RealNVP, Glow) on standard high-dimensional benchmarks? Typical examples include tabular datasets (e.g., UCI benchmarks) or image density estimation.

**Q3. Radial Flow Performance on General Distributions:**

How does the Radial Flow architecture perform on arbitrary distributions that lack an inherent radial or circular structure (e.g., checkerboard, or pixel images)?

**Limitations:**

yes

**Strengths And Weaknesses:**

**Strengths**

- **Novel Bijection Families:** The paper introduces three carefully crafted families of analytic bijections that satsifies some mathematical properties. The construction is also done in a logically grounded manner.
- **Application to Physics:** The authors demonstrate the use of their bijections in a specfic application in physics (\phi^4 lattice field theory). They also introduced a problem-specific "zero-mode bijection" to address mode collapse problem.
- **Presentation:** The paper is self-contained and is easy-to-follow, thanks to the visual explanations.

**Weaknesses**

**1. Significance: Limited Practical Impact in the Current Generative Modeling Landscape**

- **Disconnect from Mainstream Trends:** The paper focuses on discrete normalizing flows (specifically designing scalar bijections for coupling layers). However, the mainstream generative modeling community has largely shifted towards continuous normalizing flows trained via simulation-free methods (e.g., Diffusion Models, Flow Matching). The practical significance of optimizing hand-designed discrete bijections feels somewhat marginal in this current context.
- **Unclear Practical Bottleneck:** While the authors correctly identify the theoretical sub-optimalities of existing scalar bijections (e.g., affine transformations lacking expressiveness, monotonic splines being only piecewise smooth and bounded), they do not clearly demonstrate how these theoretical gaps translate into critical practical failures in real-world applications. Consequently, proposing new hand-designed bijections appears to address a relatively minor problem rather than a significant bottleneck.

**2. Soundness: Insufficient Justification for the Desiderata**

- The authors establish a strict set of desiderata for scalar bijections (global smoothness $C^\infty$, global domain $\mathbb{R}$, analytic invertibility). However, the paper lacks a compelling argument or empirical evidence showing why all these strict conditions are necessary for effective density estimation. The motivation relies heavily on mathematical elegance rather than demonstrating a critical empirical need for these properties.

**3. Soundness: Lack of Complex, Large-Scale Experimental Validation**

- **Toy-ish Benchmarks:** The authors claim their proposed bijections can serve as "drop-in replacements" for standard methods like RealNVP. However, the experiments are heavily restricted to 1D/2D toy distributions and a specific physics application ($\phi^4$ lattice field theory). There is no evaluation on standard, high-dimensional complex distribution modeling tasks (e.g., image density estimation on CIFAR-10/ImageNet, or standard UCI tabular benchmarks) where RealNVP and Spline flows are typically evaluated.
- **Inductive Bias in Radial Flow Evaluation:** The proposed Radial Flow architecture is exclusively evaluated on 2D datasets that inherently possess a radial or circular structure (e.g., 2D Spirals, Gaussian Mixture in a circle). Given the strong inductive bias of radial flows, it is entirely expected that they would excel here. The paper fails to verify whether this architecture provides any practical benefit when modeling arbitrary target distributions without such geometric structure.

---

> ### Author Rebuttal · Authors · 2026-03-31
>
> We thank the reviewer for recognizing the mathematical rigor and presentation quality.
>
> **New results since submission.** In response to reviewer feedback, we have conducted substantial additional experiments:
> - *CIFAR-10* (RealNVP architecture): sinh bijections achieve **3.37 bpd**, improving over both the published RealNVP result (3.49) and Durkan et al.'s RQ-NSF(C) (3.38), despite the latter using a different architecture.
> - *UCI tabular* (5 datasets): we reimplemented the coupling-based NSF architecture in JAX with matched hyperparameters across all bijection variants. Sinh matches spline as a drop-in replacement across all datasets. Combining spline + sinh matches or improves over pure spline, exceeding published RQ-NSF(C) on POWER (0.65 vs 0.64 ± 0.01). On MINIBOONE (3–5 seeds per variant), sinh shows notably lower variance. Full error bars from additional training seeds will be included in the revision.
> - *phi^4 (400D)*: added spline baseline. Analytic bijections consistently outperform splines (ESS ~39% vs 34%).
> - *Derivative quality*: at equal density fit (>99.9% ESS), the MSE of $\tfrac{d^2}{dx^2} \log p(x)$ is **900x lower** for analytic bijections than splines.
>
> We address each concern below.
>
> **Significance: discrete NFs and practical impact.**
> Discrete normalizing flows remain essential in several active scientific domains (e.g. simulation-based inference, quantum field theory simulations, and condensed matter physics) where exact likelihood evaluation and cheap sampling are needed simultaneously. The analytic bijections are general-purpose building blocks for normalizing flows, analogous to activation functions (GeLU, SiLU) in standard networks. Like activation functions, they are derived from principled mathematical criteria rather than learned end-to-end. They are foundational components that slot into any architecture using scalar bijections, as the new results above demonstrate.
>
> **Q1: Concrete scenario where C-infinity matters.**
> To demonstrate the practical relevance of smoothness, we trained a cubic bijection and a spline on a 1D target with roughly matched parameter counts.
> Both achieve >99.9% ESS and visually indistinguishable densities.
> When estimating $H = \tfrac{d^2}{dx^2} \log p(x)$, however, the spline's MSE is **900x larger** than that of the analytic bijection in this setup.
> This quantity enters directly in scientific settings such as condensed matter physics and information geometry.
> Therefore, smoothness of log-density derivatives is a practical requirement in these applications.
> We also observe that training-loss variance decreases substantially over time for the analytic bijection, while remaining higher for splines, suggesting a smoother optimization landscape.
> We will add this analysis in an appendix section.
>
> **Q2: Standard benchmarks.**
> See the new results above. On CIFAR-10, sinh used in the RealNVP architecture achieves 3.37 bpd, beating both the original affine result (3.49) and Durkan et al.'s RQ-NSF(C) (3.38). Across all five UCI tabular datasets, sinh matches spline performance as a drop-in replacement; combining them yields further gains (e.g. exceeding published RQ-NSF(C) on POWER). We hope these results address the requested standard benchmark evaluation.
>
> **Q3: Radial flow on non-radial distributions.**
> On the GMM target of Fig 8, which has 5 separated modes that break radial symmetry, the radial flow achieves clean results with ~1000x fewer parameters than the coupling baseline.
> Radial flows excel in low-dimensional settings, as explained in the paper's limitations.
> Their strengths are training stability (learning rates 10x higher), geometric interpretability, and parameter efficiency, making them a practical tool for O(1)-dimensional problems.
>
> We hope the new benchmark results and clarifications address the concerns raised, and that the reviewer reconsiders the score in light of these additions.

---

> > ### Author Rebuttal · Reviewer_fAER · 2026-04-02
> >
> > The additional experimental results on more complex datasets successfully validate the proposed analytic bijections, adequately addressing my previous concern. I do have one follow-up question regarding the smoothness desiderata: Table 1 states that some cubic spline variants satisfy $C^k$. Is this the same cubic spline formulation used in the main experiments and the new $H = \frac{d^2}{dx^2} \log p(x)$ estimation experiment? Why we need $C^\infty$ instead of $C^k$ for estimating H?

---

> > > ### Author Response · Authors · 2026-04-02
> > >
> > > We thank the reviewer for the opportunity to clarify this point.
> > >
> > > **Which spline is used.**
> > >
> > > Table 1 uses $C^k$ as a general label for spline smoothness.
> > > We used the rational quadratic (RQ) splines from Durkan et al. in all our experiments, including the $H$ estimation, which is the $k=1$ case: values and first derivatives match at knots, but higher derivatives do not.
> > > RQ splines are the most widely adopted spline bijection in the normalizing flow literature, to our knowledge.
> > >
> > > **Why smoothness matters for $H$.**
> > >
> > > $C^\infty$ is not strictly *required* to compute $H$ — knots are isolated points, so $H$ exists almost everywhere even for $C^1$ splines.
> > > The finding is empirical: the spline's MSE on $H$ is ~900x larger, despite both methods achieving >99.9% ESS (nearly identical density fits).
> > >
> > > The intuition is that a $C^1$ spline is stitched together from independent pieces that only agree on values and slopes at boundaries.
> > > Training samples in one bin do not constrain higher-order behavior (e.g. the curvature entering $H$) in neighboring bins. This information is lost at each knot.
> > > By contrast, an analytic bijection is globally determined: fitting the density well anywhere implicitly constrains all derivatives everywhere.
> > > The 900x gap is the empirical consequence. Splines can match the density ($p$) well, yet still poorly approximate its second derivatives ($H$), and stacking coupling layers would compound this effect.
> > >
> > > **$C^k$ vs $C^\infty$.**
> > >
> > > Higher $k$ adds more constraints at knot boundaries, which would help, but neighboring pieces are coupled only through finitely many conditions at each knot, not through a global constraint.
> > > Even matched derivatives can exhibit sharp features: for $C^1$ splines, $f'$ is continuous but can have sharp kinks at knots, and we observe that already the first derivative of $\log p$ shows noticeable spikes.
> > > $C^\infty$ analytic bijections sidestep this entirely: the function is globally smooth, so $\log p$ inherits smoothness at all orders and no downstream quantity (e.g. Fisher information, HMC trajectories) suffers from bin-boundary effects.
> > > This is consistent with our observation that training-loss variance also decreases for analytic bijections, suggesting a smoother optimization landscape overall.
> > >
> > > To clarify the role of this experiment: the $H$ comparison is a concrete demonstration that smoothness of the bijection has observable downstream consequences — answering the reviewer's original Q1.
> > > These consequences show up directly in gradient-derived quantities that matter in certain applications, and appear to also affect training smoothness (lower loss variance over time).
> > > A higher-$k$ spline would likely reduce the gap, but the qualitative point stands: any piecewise construction limits information sharing across bins, and $C^\infty$ analytic bijections avoid this by construction.

---

### Official Review · Reviewer_61VC · 2026-03-13

**Soundness:** 3
**Presentation:** 3
**Significance:** 2
**Originality:** 3
**Overall Recommendation:** 4
**Confidence:** 4

**Summary:**

This paper proposes a new class of nonlinear scalar bijections for normalizing flows which are smooth and have closed-form inverses, properties which do not all appear in previous bijections (e.g. affine, splines, residual nets). These bijections are used a drop-in replacements for instances of coupling flows. Moreover, the paper introduces a new normalizing flow architecture, radial flows, which bijections only depend on the radius relative to its center and can be interpretable in settings with high radially symmetry. Both qualitative and quantitative results are presented for 1D, 2D toy distributions and lattice-field theory application of dimension of 400.

**Compliance With Llm Reviewing Policy:**

Affirmed.

**Final Justification:**

The authors have added substantial experiments and addressed my concerns. Even though the evaluations are mostly toy cases, I am more convinced that their new parameterizations of bijections will be used by others working on normalizing flows.

**Key Questions For Authors:**

- What is the run-time and scalability of your method compared to splines? I imagine that inversion for splines is more costly in general, so it could make the results more compelling to include run-time analysis of your methods vs baselines, along with the prediction performance.
- Is there a reason why splines are not also compared for the lattice-field theory experiments?
- How feasible is it for spherical harmonics to scale radial flows to high dimensions?
- Can the authors provide the complete quantitative and qualitative results mentioned above in weaknesses?

**Limitations:**

yes

**Strengths And Weaknesses:**

Strengths

- It is useful to delineate the shortcoming of existing bijections used in normalizing flows. A new class of scalar bijections are identified with analytical inverse and are simple to implement.
- The radial flow architecture is novel to the best of my knowledge and the interpretability with regard to angle-dependent radial bijections. I suppose it could be useful to some scientific applications or settings with high radial symmetry.

Weaknesses

- The most novel contribution Radial flows are only presented for 1D and 2D problems. The authors state that it can be extended to high-dimensions with spherical harmonics, however it is known to be a somewhat costly to incorporate the spherical harmonics for high dimension without heavy truncation. It would have been interesting to see that tradeoff.
- The paper claims there are many theoretical benefits of having C-infinity bijections, but the results don't show any real practical benefit over splines, which appears to be a very strong baseline in Figure 3.
- The qualitative comparison feels incomplete in figure 6 because it compares radial flows to the Real NVP type affine flows, but not splines even. However, splines clearly dominate affine bijections in the quantitative figure 9.
- There seems to be no quantitative comparison of radial flows, only qualitative. It would be nice to see a metric for radial flows compared to coupled flows on the spiral 2D.

---

> ### Author Rebuttal · Authors · 2026-03-31
>
> We thank the reviewer for the constructive feedback, particularly the identification of missing comparisons.
>
> **New results since submission.** In response to reviewer feedback, we have conducted substantial additional experiments:
> - *CIFAR-10* (RealNVP architecture): sinh bijections achieve **3.37 bpd**, improving over both the published RealNVP result (3.49) and Durkan et al.'s RQ-NSF(C) (3.38), despite the latter using a different architecture.
> - *UCI tabular* (5 datasets): we reimplemented the coupling-based NSF architecture in JAX with matched hyperparameters across all bijection variants. Sinh matches spline as a drop-in replacement across all datasets. Combining spline + sinh matches or improves over pure spline, exceeding published RQ-NSF(C) on POWER (0.65 vs 0.64 ± 0.01). On MINIBOONE (3–5 seeds per variant), sinh shows notably lower variance. Full error bars from additional training seeds will be included in the revision.
> - *phi^4 (400D)*: added spline baseline. Analytic bijections consistently outperform splines (ESS ~39% vs 34%).
> - *Derivative quality*: at equal density fit (>99.9% ESS), the MSE of $\tfrac{d^2}{dx^2} \log p(x)$, is **900x lower** for analytic bijections than splines.
>
> We address each point below.
>
> **Practical benefit over splines.**
> As shown in the new results: analytic bijections outperform splines on phi^4 (ESS 39% vs 34%), CIFAR-10 (3.37 vs published 3.38 bpd), and in derivative quality (900x lower MSE). Across all five UCI datasets, sinh matches spline as a drop-in replacement; combining them captures complementary structure (exceeding published RQ-NSF(C) on POWER). The practical benefit is real and consistent across physics, image, and tabular benchmarks.
>
> **Splines in phi^4 experiments**
> This was an oversight: the original submission focused on the affine baseline from the Albergo et al. reference architecture. Splines are now included and fall between affine and analytic bijections.
>
> **Fig 6: missing spline comparison.**
> Fig 6 compares coupling *architecture* (axis-aligned transforms causing folding) vs radial *architecture* (radial transforms, no folding). It is not a scalar bijection comparison. Splines in the coupling layer would exhibit the same folding artifacts because the issue is the coupling structure, not bijection expressivity. We are happy to add a spline-coupling row to the appendix that shows this.
>
> **Quantitative radial flow metrics.**
> On the 2D spiral (Fig 6), the radial flow achieves NLL −0.79 vs −0.52 for the RealNVP coupling baseline, with ~1000x fewer parameters. The comparison in Fig 6 is primarily about architectural properties (folding artifacts, geometric structure), but the radial flow also fits the density better on this target.
>
> **Runtime and scalability.**
> We find that linearly chaining operations, as in copies of scalar bijections, is generally more expensive than adding more knots to a spline bijection (chaining multiple spline bijections, on the other hand, would similarly increase the cost).
>
> **Spherical harmonics scalability.**
> To clarify: both angle-independent and angle-dependent radial layers are algorithmically well-defined in any dimension.
> For angle-dependent layers, any neural network taking the unit vector x/|x| as input works immediately with no spherical harmonics needed.
> The explicit Fourier/spherical harmonic expansion discussed in the paper is limited to 2D (one angle), but this is a choice of interpretable parameterization, not a limitation of the radial architecture itself.
> That said, we do expect radial flows are best suited as a low-dimensional tool, where their training stability, interpretability, and parameter efficiency are most valuable.
>
> We hope these results and clarifications address the concerns raised, and that the reviewer reconsiders the score in light of these additions.

---

> > ### Author Rebuttal · Reviewer_61VC · 2026-04-03
> >
> > Thank you for the detailed response.  I believe the new results will significantly strengthen the paper. I will consider this when adjusting my final rating accordingly.

---

> > > ### Author Response · Authors · 2026-04-06
> > >
> > > We thank the reviewer for the positive acknowledgement and for the constructive review process. We hope the reviewer will update their score as they see fit.

---

### Decision · Program_Chairs · 2026-04-30

**Decision:**

Accept (regular)

**Comment:**

After the discussion, the reviewers and I are in agreement that the paper should be accepted. The new analytic bijections seem useful for the community, the statements in the paper seem sound, and the experimental results and baselines are reasonable. There was some discussion on properly comparing to other bijections (especially splines), and the authors have provided additional results on CIFAR-10, UCI, and phi^4 settings